# Characterization of Transcriptome Dynamics during Early Fruit Development in Olive (*Olea europaea* L.)

**DOI:** 10.3390/ijms24020961

**Published:** 2023-01-04

**Authors:** Maria C. Camarero, Beatriz Briegas, Jorge Corbacho, Juana Labrador, Mercedes Gallardo, Maria C. Gomez-Jimenez

**Affiliations:** 1Laboratory of Plant Physiology, University of Extremadura, Avda de Elvas s/n, 06006 Badajoz, Spain; 2Laboratory of Plant Physiology, University of Vigo, Campus Lagoas-Marcosende s/n, 36310 Vigo, Spain

**Keywords:** cell division, cell expansion, flow cytometry, fruit growth, hormone, olive, ploidy, transcriptome

## Abstract

In the olive (*Olea europaea* L.), an economically leading oil crop worldwide, fruit size and yield are determined by the early stages of fruit development. However, few detailed analyses of this stage of fruit development are available. This study offers an extensive characterization of the various processes involved in early olive fruit growth (cell division, cell cycle regulation, and cell expansion). For this, cytological, hormonal, and transcriptional changes characterizing the phases of early fruit development were analyzed in olive fruit of the cv. ‘Picual’. First, the surface area and mitotic activity (by flow cytometry) of fruit cells were investigated during early olive fruit development, from 0 to 42 days post-anthesis (DPA). The results demonstrate that the cell division phase extends up to 21 DPA, during which the maximal proportion of 4C cells in olive fruits was reached at 14 DPA, indicating that intensive cell division was activated in olive fruits at that time. Subsequently, fruit cell expansion lasted as long as 3 weeks more before endocarp lignification. Finally, the molecular mechanisms controlling the early fruit development were investigated by analyzing the transcriptome of olive flowers at anthesis (fruit set) as well as olive fruits at 14 DPA (cell division phase) and at 28 DPA (cell expansion phase). Sequential induction of the cell cycle regulating genes is associated with the upregulation of genes involved in cell wall remodeling and ion fluxes, and with a shift in plant hormone metabolism and signaling genes during early olive fruit development. This occurs together with transcriptional activity of subtilisin-like protease proteins together with transcription factors potentially involved in early fruit growth signaling. This gene expression profile, together with hormonal regulators, offers new insights for understanding the processes that regulate cell division and expansion, and ultimately fruit yield and olive size.

## 1. Introduction

The olive tree (*Olea europaea* L., Oleacea) produces fruit that rank among the world’s leading oil crops [1]. Although many authors have described polyploidy [2,3], the species is diploid (2n = 2X = 46). Its genome size is approximately 1800 Mb [4]. Though it is technically a fleshy drupe from hermaphroditic flowers, the fruit is commonly called a ‘stone fruit’ because the seed coat is enclosed in a stone-hard, lignified endocarp [5]. The olive fruit is composed of two parts: the flesh, which is of maternal origin, and the embryo, which has different genetic origins that explain certain characteristics of the oil composition in different environments [6].

The size increase of the olive fruit depends on two distinct mechanisms: (1) young fruit growth (cell division and expansion) before endocarp lignification, and subsequently, (2) fruit pulp growth (cell expansion) after endocarp lignification (late fruit growth). As the young olive fruit accumulates water, its cells swell and also rapidly proliferate. This early developmental feature of the fruit becomes patent some 50 days following ovary fertilization, as most of the nutritional resources become mobilized at this early stage while the other parameters tend to slow down [5,6]. After this early stage and until fruit ripening, fruit pulp growth is due solely to cell expansion, and oil starts to accumulate. The oil content and composition are determined by the cultivar with little environmental influence, whereas other compounds (phenols and sterols) depend more on the environment [6]. Because of the increasing oil content during fruit pulp growth, which can reach 30% (fresh weight) at full ripening [6], and due to the high commercial value of the oil, most studies to date have attempted to elucidate the molecular bases of olive fruit ripening [7,8,9,10,11,12,13,14,15,16,17,18,19,20]; however, to date, few studies have shown candidate genes associated with early olive fruit development [21,22,23,24,25,26,27,28]. In particular, the transcriptomic control of changes associated with early fruit development in olive remains unknown. 

In fleshy fruit, early development requires a precise orchestration of cell division, cell expansion, and cell differentiation by integrating endogenous signals and various environmental cues [29,30,31,32,33,34]. Fruit growth begins immediately after the ovary is pollinated, triggering vigorous cell division. This cell division is the primary driver for growth within the ovary. Spatial changes in cell division rate or duration within the ovary thus holds sway over the final fruit size [34]. In tomato (*Solanum lycopersicum*), a model system for fleshy fruit development [34,35], the prime factors controlling ultimate tomato fruit size are cell division and cell expansion [30,31,36]. Moreover, in many species, including tomato, the transition from cell division to expansion phases is accompanied by endoreduplication [31,32,33,37,38]. Because cell size for a certain cell type is generally proportional to the quantity of nuclear DNA, an effective strategy of cell growth is endoreduplication, which often occurs in differentiated cells that are large or metabolically highly active [39,40,41]. An understanding of the interaction involving endoreduplication, cell division, and cell expansion processes is vital for predicting the appearance of key morphological traits (fruit size, shape, mass, and texture). In olive, however, the ploidy levels during early fruit development have not yet been explored, and several questions remain regarding not only the rate and duration of cell division during early fruit development but also its relative importance on the final fruit size. According to previous studies analyzing the olive fruit histology by qualitative (cross-section) methods, cell division and cell expansion occur concomitantly in the first phase of fruit growth [42,43,44]. However, no available studies have investigated the mitotic activity of olive fruits by flow cytometry during early fruit development. Recently, it has been demonstrated that under water stress during the fruit pulp growth (cell expansion) in olive, even though olive fruit growth appears to stop, stress-induced cell division occurs, and the final size of the olive fruit is not severely diminished when the trees are irrigated again during the period of fruit pulp growth [45]. 

In recent years, notable progress has been made in understanding the molecular mechanisms that coordinate the interaction of transcriptional control and hormonal activity in early fruit development [38,46,47,48,49,50,51,52,53,54]. The modification of genes that act in hormonal regulation can alter fruit size, given that hormone contents can undergo major shifts during fruit growth [34]. In fact, using mutants or exogenous treatments, many studies have indicated the decisive role of hormones for early fruit-growth regulation [55,56,57,58,59]. In olive, high levels of cytokinins (CKs), gibberellins (GAs), and polyamines (PAs) have been reported in the developing fruits [27,60,61]. Moreover, the application of exogenous brassinosteroid (BR) promoted early fruit growth in olive, whereas blocking BR synthesis with brassinazole (Brz) slowed down the fruit growth rate [62]. Likewise, the data have provided new findings on the role of BRs in modulating the composition and gene expression of sterols and sphingolipids, these being lipophilic membrane components essential for cellular functions during early olive fruit growth [62,63]. This latter study also revealed the upregulation of *β*-sitosterol biosynthesis by BR at the transcriptional level during early olive fruit growth. In addition, previous data have demonstrated that free endogenous PAs may regulate olive flower anthesis mainly through S-adenosyl methionine decarboxylase (SAMDC) enzyme activity and an expression gene localized in the ovary. The same study has indicated that PAs appear to correlate positively with cell division during early fruit development in olive [27]. Nevertheless, knowledge remains scant on the hormone content and composition of olive fruits during early fruit development.

The purpose of the present study was to gain a more complete picture of gene expression and hormonal control during early fruit development in olive. For this, the cytology, ploidy, and transcriptome dynamics associated with early fruit development in combination with data on hormonal content were analyzed. The cell area and mitotic activity of the developing olive fruits were first investigated by flow-cytometric analysis in in order to characterize the duration of the cell division and expansion phases. Next, transcriptional and hormonal changes in olive fruit were investigated during early fruit development as a means of establishing candidate genes associated with distinct phases of early development. Specific potential genes and their associated hormonal pathways were determined to participate in the phases of cell division and expansion during early olive fruit development.

## 2. Results and Discussion

### 2.1. Morphological and Cytological Changes during Early Fruit Development in Olive

The early fruit development in olive was explored by monitoring weight, longitudinal and transverse diameters, and size of cells in an olive cultivar ‘Picual’ with large sized and elongated fruits. As shown in Figure 1, the olive fruit, after ovule fertilization (concomitant with anthesis), showed a highly reproducible growth progression in terms of increases in weight, longitudinal and transverse diameters, index shape, and growth rate up to 42 DPA, followed by a period of endocarp lignification for fruit under the experimental conditions used (Figure 1A–F). Immediately following fertilization, the developing fruits grew more in length than in width (Figure 1C,D), resulting in an increase in the fruit-shape index (Figure 1E). This trend is characteristic of the elongated fruit shape of the ‘Picual’ olive, and consequently its fruit size (weight) was augmented 128-fold (±6.0) from anthesis to 42 DPA. Greater cell size was apparent in both the epicarp and mesocarp (pericarp) cells of the fruit during the early stages (0–42 DPA) (Figure 1H,I). In fact, the epicarp- and mesocarp-cell area in the pericarp of one ovary at anthesis increased 5- and 13-fold, respectively, in the developing fruit at 42 DPA. Pericarp cells expanded at similar rates (cell area/day) for 21 DPA. From 21 to 28 DPA, cell expansion in the pericarp was dramatic, with the cell expansion rate doubling up to 42 DPA. Variations in pericarp thickness were linked mainly to variations in mesocarp cell size (Figure 1G–I). Thus, the 128-fold increase in olive fruit weight up to 42 DPA was related to a 7-fold increase in longitudinal diameter, a 5-fold increase in transverse diameter, and a 13-fold increase in mesocarp cell size.

### 2.2. Ploidy Level during Early Fruit Development in Olive 

Here, for the first time, detailed quantitative data is reported at the ploidy level in olive fruit throughout early development. Cell division during early olive fruit development was characterized by flow cytometric analysis of the nuclear-DNA contents from 0 to 42 DPA (Figure 2). At anthesis (0 DPA) and the 7 DPA stage, most of the nuclei were 2C (88% and 65.2% of total nuclei, respectively) and only a low proportion of endoreduplicated nuclei >4C was found (0.8 % and 1.3 % of total nuclei; respectively, Figure 2). 

From 0 to 14 DPA, the proportion of 4C cells increased with time, while the proportion of 2C cells decreased in olive fruits. At 14 DPA, an early stage of fruit development, 4C cells represented the highest proportion of cells in the olive fruits (63.2%), indicating intensive cell division, while the proportion of 8C cells increased in comparison to those in fruits at 7 DPA (1.3% and 15.8% of total nuclei at 7 and 14 DPA, respectively, Figure 2). By contrast, at 28 DPA, cell division was not activated in the fruits based on the increased ratio of 2C to 4C DNA levels relative to that at 21 DPA (Figure 2). The 2C cells represented 40.3% and 59.4% of the cells, and the 4C cells represented 57.6% and 40.4% of the cells in the fruits at 21 and 28 DPA, respectively. The 8C cells represented 2.1% and 0.3% of the cells in the fruits at 21 and 28 DPA, respectively. From 28 to 42 DPA, most of the nuclei were 2C (59.39, 89.3%, and 93.5% of total nuclei of fruits at 28, 35, and 42 DPA, respectively), while only low proportions of 4C and 8C nuclei were detected in these fruits (Figure 2). Thus, the results show that the maximal proportion of 4C cells, which is an indirect estimate for cell division, in olive fruits was reached at 14 DPA, indicating that intense cell division was activated in the fruits at this time. 

In olive cultivars, longer durations of the cell division phase have been qualitatively reported through 8 to 10 weeks [42,43,44]. However, in the present study, flow cytometric analysis revealed that cell division was triggered by pollination in the ‘Picual’ olive fruit from 0 to 21 DPA (Figure 2). The data here offered no evidence for cell division in fruits from 28 to 42 DPA, and the increased fruit size (weight) in this period apparently resulted from cell expansion, while both cell division and expansion coexisted at gradually increasing rates in the fruits until 21 DPA (Figure 1 and Figure 2). A greater pericarp cell size was evident both in dividing cells (0–21 DPA) and in post-mitotic expanding cells (28–42 DPA) in the ‘Picual’ olive fruit, which notably contributed to significantly enlarged olive fruit at all the stages examined (Figure 1).

Particularly evident is the transition between 14 and 28 DPA, which corresponds to the shift from intense cell division to cell division arrest in ‘Picual’ olive fruit. As expected, after the cessation of cell division, pericarp cell size most rapidly increased in ‘Picual’ olive fruit, and the maximum relative rate of cell expansion in the fruit occurred at 42 DPA (Figure 1); the maximum relative rate of cell division in the fruit was found at 14 DPA (Figure 2). At this stage, a low but significant proportion of endoreduplicated cells of up to 8C (one endocycle) was detected in olive fruits compared with tomato fruits (up to 256C or even 512C) [31,33,64,65,66]. Endoreduplication increases ploidy in individual cells and reportedly correlates not only with high metabolic activities, cell differentiation, post-mitotic cell growth, and rapid anisotropic cell expansion, but also with the capacity to react under DNA damage [67,68]. In particular, the ploidy increase is strongly correlated with increased cell size [33,67]. In tomato fruit, previous studies have reported that endoreduplication begins in developing ovaries when organogenesis-related cell division ends [31,33]. However, with the exception of some Rosaceae species (e.g., apricot, peach, and plum), endoreduplication does not occur in most of the species where fruit development lasts for a long period (over 14 weeks) of time [66], as is the case of the olive fruit.

### 2.3. Overall Transcriptional Changes during Early Fruit Development in Olive

Based on the cytological and ploidy analyses of early fruit development (Figure 1 and Figure 2), to ascertain the molecular mechanism of early fruit growth in olive, a comparison was made of the transcriptome, using RNA-seq, of whole ‘Picual’ olive fruit at selected stages of early fruit development: 0 DPA (P0, anthesis, fruit set), 14 DPA (P14, cell division, mostly dividing cells), and 28 DPA (P28, post-mitotic cell expansion, mostly expanding cells). The use of these three samples (P0, P14, and P28) of developing ‘Picual’ olive fruit, which represent critical physiological changes during early fruit development, allowed the preferential identification of post-mitotic cell expansion-related genes (Figure 3A). 

The RNA-Seq analysis averaged 40 million reads per sample, of which 78% mapped to the 132,819 annotated transcripts from the *Olea europaea* var. *sylvestris* reference genome [69] (Appendix A). From the three olive samples, a PCA was performed using transformed read counts. The three biological replicates for each sample clustered together, thus indicating that the expression levels among replicated samples were closely associated (Appendix A). A total of 24,168 differentially expressed genes (DEGs) related to early fruit development in olive were identified (Figure 3B; Appendix A). 

Among the 24,168 DEGs, 16,943 genes were developmentally responsive in the first comparison (between the fruit at 14 DPA and 0 DPA, P14 vs. P0), and 7225 were developmentally responsive in the second comparison (between fruit at 28 DPA and 14 DPA, P28 vs. P14) (Figure 3B; Appendix A). In the first comparison (P14 vs. P0), 8040 genes were upregulated, and 8903 were downregulated in fruit at 14 DPA. In the second comparison (P28 vs. P14), 3443 genes were upregulated, and 3782 were downregulated in fruit at 28 DPA (Figure 3B; Appendix A). A comparison of the genes that were developmentally responsive during early fruit development indicated that 1461 were upregulated in both comparisons, and that 3691 were downregulated in both comparisons (Figure 3C; Appendix A). Thus, most of the 24,168 DEGs display a distinct temporal expression consistent with the succession of the different phases of early fruit development (Figure 3C; Appendix A). 

Additionally, 162 transcripts that were identified could be considered as fruit-set specific (specific genes at anthesis) among the transcripts expressed in the fruit at 0 DPA (fruit set) since they were not detected in the other fruit development stages analyzed (Appendix A; Figure 3D), including homologues of stress, cell wall, transport, cell division, transcription factor (TF), and hormone-related genes. An appreciable proportion (almost 30%) of them encode proteins with unknown functions or present no homology with any known genes. Similarly, among the transcripts expressed in the olive fruit at 14 DPA (mostly dividing cells), 67 transcripts could be considered to be cell division specific (specific genes at 14 DPA) (Appendix A; Figure 3D). These mainly included genes involved in cell growth processes such as cell wall synthesis and modification, sugar and organic acid transport and metabolism, and hormonal metabolism and signaling. Among the transcripts expressed in the fruit at 28 DPA (post-mitotic cell expansion), only four transcripts (*CS2*, *DIR15, EXPB2* and *cox-6A*) could be considered to be cell expansion-specific (specific genes at 28 DPA) (Appendix A; Figure 3D): (1) *CS2* coding for chorismate synthase 2 protein controls a key step in the shikimate pathway and catalyzes the transformation of 5-enolpyruvylshikimate 3-phosphate to chorismate, which serves as the initiator metabolite for the synthesis of aromatic amino acids and secondary metabolites (the pathway ‘phenylalanine, tyrosine, and tryptophan biosynthesis’ and other pathways) [70]; (2) *DIR15* coding for dirigent protein 15-like protein, which is involved in cell wall metabolism [71]; (3) *EXPB2* coding for the expansion of B2 protein, which is involved in cell expansion and other developmental events during which cell wall modification occurs [72]; finally, (4) *cox-6A* coding for cytochrome c oxidase subunit 6a protein is involved in the pathway oxidative phosphorylation, which is part of energy metabolism [73]. Moreover, for validation of the RNA-seq results, a qRT-PCR was performed to determine the levels of *OeCS2*, *OeDIR15*, *OeEXPB2,* and *Oecox-6A* expression across the three samples, which exclusively showed a significant expression level in the fruit at 28 DPA (Appendix A). Thus, they were considered to be prime candidates for further molecular research on the programs of cell division and expansion in olive fruit and their regulation.

### 2.4. Gene Ontology Functional Enrichment Analysis of Differentially Expressed Genes

For an overall view concerning the functions and processes altered during early olive fruit development, the DEGs were classified using the Gene Ontology (GO) database. Furthermore, GO accessions were assigned to the DEGs based on similar sequences in known proteins available in the UniProt database in addition to InterPro as well as the Pfam domains that these contain. The GO terms ‘Transcription’, ‘ATP binding’, and ‘Integral component of membrane’ were most represented among the categories of biological processes (Figure 4; Appendix A), molecular functions (Appendix A), and cell components (Appendix A), respectively. 

Within the ‘Biological process’ category, the over-represented group in the dividing olive fruit (14 DPA) with the greatest number among the DEGs was ‘Regulation of transcription’, ‘Translation’, ‘Protein ubiquitination’, ‘Defense response’, ‘Protein transport’, ‘Protein autophosphorylation’, ‘Carbohydrate metabolic process’, ‘Cell wall organization’, ‘Response to salt stress’, ‘Transmembrane transport’, ‘Hormone-mediated signaling pathway’, and ‘cell division’ (Figure 4A). Remarkably, the expanding olive fruit (28 DPA) also bore a significant representation of transcripts associated with ‘Regulation of transcription’, ‘Translation’, ‘Protein ubiquitination’, ‘Protein autophosphorylation’, ‘mRNA processing’, ‘Protein transport’, ‘Cell division’, ‘Carbohydrate metabolic process’, and ‘Cell wall organization’ (Figure 4C), indicating that the same biological processes might necessitate different gene sets in different phases during early fruit development in order to support their activities. 

### 2.5. Characterization of Cell Cycle-Related Genes Associated with Early Fruit Development in Olive

An examination of the current data set corroborates the development-induced accumulation of transcripts presumably participating in the basic cell cycle machinery, including DEGs that encode cyclins (CYC, 48 genes), cyclin-dependent kinases (CDKs, 8 genes), CDK inhibitor proteins (CKIs, 10 genes), homologues of the retinoblastoma (RB) protein (2 genes), and the E2F TFs (4 genes) during early olive fruit development (Figure 5 and Figure 6; Appendix A). 

CYCs, the regulatory subunits of their respective CDKs, constitute major components in the cell cycle progression machinery [34,74]. In fact, certain cyclins are discontinuously expressed over the cell cycle, as their synthesis as well as their degradation prove to be strictly programmed [75]. This raises the question as to whether the intense cell division phase during early olive fruit development involves the utilization of specific cyclins. A total of 48 cyclins were included in the data set and were retained for further phylogenetic analysis, of which 5 are A-type, 4 are B-type, 2 are C-type, 16 are D-type, 2 are L-type, and 2 are U-type cyclins (Figure 5C; Appendix A). Among the 48 genes, 16 (3 A-type, 1 B-type, 6 D-type, 1 L-type, and 1 U-type cyclins) were upregulated exclusively in olive fruit at 14 DPA compared with 0 DPA (P14 vs. P0 comparison), and 5 (one C-type, 3 D-type, and one U-type cyclins) were upregulated only in olive fruit at 28 DPA compared with 14 DPA (P28 vs. P14 comparison), whereas 16 genes (2 A-type, 3 B-type, 1 C-type, 7 D-type, 1 L-type cyclins, and 2 cyclin 2-like) were upregulated in both comparisons (Figure 5A,B). Among the latter, the most abundant proved to be CYCD3;1 and a CYCD6;1 proteins.

Additionally, the present data demonstrated that 11 cyclins genes (1 B-type, 1 D-type, 1 L-type, 1 P-type, 3 T-type, and 4 U-type cyclins) were downregulated in the olive fruit at 14 DPA compared with 0 DPA, and only 1 cyclin gene (D-type) was exclusively downregulated in the olive fruit at 28 DPA compared with 14 DPA, whereas 2 (one C-type and one L-type cyclin) were downregulated exclusively in olive fruit at 28 DPA compared with 14 DPA (P28 vs. P14 comparison) (Figure 5; Appendix A). Moreover, differences in expression among different A-type (A1 vs. A2 and A3) cyclins and different B-type (B2 vs. B3) cyclins during early fruit development make it possible to discriminate olive fruits that have an arrested cell division phase. Expressions of A2, A3, B3, and U1 cyclins in olive fruit are restricted to intensive cell division activity during early fruit development. For further analysis of CYC family genes, a qRT-PCR analysis was conducted to assess the expression of eight members of this family in fruits at 0, 14, and 28 DPA (Figure 5D). The qRT-PCR test in the dividing olive fruit corroborated the enrichment in A2, A3, B3, and U1 cyclin genes, as well as the enrichment of A1, B2, and C1 cyclin genes in the expanding olive fruit (Figure 5D). The hypothesis is that the differences in time of expression of cyclin A2 and A3 versus cyclin A1 as well as of cyclin B3 versus cyclin B2 may reflect specific functions during olive fruit development, including A-, B-, and D-type CYCs.

Similarly, only one member of the CDK family (*CDKC1*) was upregulated exclusively in olive fruit at 14 DPA compared with 0 DPA, whereas six of the eight members of the CDK family were downregulated exclusively in olive fruit at 14 DPA compared with 0 DPA (P14 vs. P0 comparison) (Figure 6A; Appendix A). Of particular interest is one member of the CDK family, homologous to CDKB1 [76], which proved to be upregulated in both comparisons (Figure 6A; Appendix A). In Arabidopsis, the M-specific CDKB1;1 constitutes the plausible candidate kinase that forms part of the mitosis-inducing factor (MIF) when it is bound to the A-type cyclin CYCA2;3, which is capable of inhibiting endoreduplication when it is fully active [77]. In the present study, after fertilization, the cell division phase results mainly from the gene activity of CYCA2, CYCA3, CYCB3, CYCU1, and CDKC1. The post-mitotic cell expansion phase is associated with the gene activity of CYCA1, CYCB2, and CYCC1, whereas transcripts encoding CYCD3;1, CYCD6;1, and CDKB1 proteins are associated with both cell division and cell expansion phases during early fruit growth in olive, in accordance with previous studies [38,47,50,51,57,78]. Therefore, the present data indicate that specific CYC and CDKs are involved in different phases during early olive fruit development.

Regulation of the cell cycle and endocycle by TFs proves to be upstream of the cyclins and CDKs. Furthermore, E2F, OBP1 (DOF TF), and THREE MYB REPEAT (MYB3R) TFs have been found to be primary regulators of the cell cycle [79]. Here, differential expression patterns of other cell cycle-related genes were investigated during early olive fruit development, including *RB*, *E2F*, *MYB3R*, and *SCARECROW-LIKE* (*SCL*) (Figure 6A, Appendix A), of which 10 genes were randomly examined and their expression patterns confirmed using qRT-PCR (Figure 6B). E2Fs are known to target genes involved in DNA repair and chromatin dynamics at the transition between the G1 and S phase. In Arabidopsis, E2FA, a protein involved in cell cycle regulation, exerts a dual function, not only maintaining cell proliferation but also stimulating the cell expansion needed in differentiating cells for the growth of organs by promoting endoreduplication [80]. In the present analysis, two members of the E2F TF family homologous to E2FA and E2FE were found to be upregulated in olive fruit at 14 DPA (P14 vs. P0), whereas one other gene homologous to E2FE was upregulated in both comparisons during early olive fruit development (Figure 6; Appendix A). In addition, the cyclin D family has a major part in the regulation of the RBR/E2F pathway, activating CDK-mediated RBR phosphorylation and upsetting its interactions with E2F in Arabidopsis [74,81,82]. The results shown here indicate a correlation between the expression of members of the cyclin D family, such as *CYCD3;1* and *CYCD6;1*, and a number of genes, such as *RBR3, E2FA*, and *E2FE* (Figure 6A; Appendix A), which could affect the balance of mitotic and endoreduplicating cells or the number of endocycles during early olive fruit growth.

By binding to and inhibiting cyclin-dependent kinase complexes, CKIs promote sustained endoreduplication in cells [67]. In addition, one gene homologous to CKI3 and two genes homologous to SIAMESE-RELATED 6 (SMR6) were exclusively transcribed in the dividing olive fruit at 14 DPA, whereas one gene homologous to SMR9 was upregulated in both comparisons. In contrast, 5 of the 10 members of the CKI family were downregulated exclusively in olive fruit at 14 DPA compared with 0 DPA, whereas only one *CKI7* gene was downregulated in both comparisons during early fruit development (Figure 6A, Appendix A). Notably, it was found that one gene homologous to AP2-like ethylene-responsive type TF (*SMOS1*), a putative orthologue of the AtSMOS1 (At2g41710) protein in Arabidopsis, is transcriptionally induced during early olive fruit development (Appendix A). SMOS1 is involved in the transcription activation of a specific set of SMR family genes, encoding plant-specific CKI and thus inhibiting cell cycle progression at G2 and promoting the onset of endoreplication [67]. In the analysis, *SMR9* exhibited a similar expression pattern as that of *SMOS1* (Figure 6A, Appendix A). The transcript level of *SMR9* rose in both comparisons during early olive fruit development, leading to the hypothesis that SMOS1/SMR9 is involved in cell cycle control, quickly triggering the repression of cell division during early olive fruit growth. Recently, [83] documented that AtSMOS1 forms a dimer with SCARECROW-LIKE28 (SCL28), a GRAS TF that in Arabidopsis, plays a critical part in regulating cell size as part of a transcriptional network downstream of the central MYB3Rs that regulate the G2 to M phase of cell cycle transition. In the present work, the early fruit growth in olive was found to be associated with upregulated *SCL28* and *MYB3R-1* transcripts (Figure 6A). This suggests that *SCL28* controls cell expansion and differentiation by promoting endoreplication onset during early fruit growth. Thus, the expression of *SCL28*, *SMOS1*, and *MYB3R-1* shows to positively correlated with early fruit growth in olive, indicating that the regulation of cell division and expansion in the olive fruit can be similar to that in other systems.

### 2.6. Differing Hormonal Composition and Gene Expression Patterns during Early Fruit Development in Olive

According to the previous data, GO enrichment identified hormone-mediated signaling pathways that may be key during early olive fruit development (Figure 4). Firstly, in the investigation of a potential relationship between early fruit development and hormonal composition, the hormone profiles of indole-3-acetic acid (IAA), GAs (GA_1_ and GA_4_), CKs [*trans*-Zeatin (*t*Z)], abscisic acid (ABA), jasmonic acid (JA), and salicylic acid (SA) were examined during early olive fruit development. As shown in Figure 7, all detected hormones decreased during the first 14 days of fruit development in olive fruit, except *t*Z, SA, and IAA. In particular, the results reveal that the *t*Z levels rose during the first 14 days of fruit development, only to disappear later from 14 to 28 DPA in olive fruit. SA levels also rose from the anthesis stage to fruit at 14 DPA, and later fell by 46% in fruit at 28 DPA, while IAA levels rose during early fruit development, reaching a maximum at 28 DPA.

The hormone profiling revealed that the olive flower at anthesis stage (fruit set) is characterized by high levels of JA and GA_1_ (Figure 7). In fact, JA was the most abundant in olive flower at anthesis, but JA levels sharply fell during early fruit development (Figure 7). Similarly, upon fertilization, a decrease of GA_4_ as well as ABA levels accompanied by a loss of GA_1_ were detected in olive fruit at 14 DPA. By contrast, at 28 DPA, the fruit showed the highest peak of hormones with a rise in IAA, GA_1_, GA_4_, and ABA levels from 14 to 28 DPA, while the *t*Z content was undetected in the fruit at 28 DPA. Notably, GA_1_ was absent from the fruit at 14 DPA but was enriched at 28 DPA (Figure 7), suggesting a specific association between this hormone and the cell expansion phase in olive fruit development. Hence, *t*Z, SA, IAA, GA_1_, GA_4_, and ABA were found to be related to early olive fruit growth. More specifically, this study indicates that, upon fertilization, the olive fruits grow as cell division prevails over expansion associated with high levels of *t*Z and low levels of GA_4_ (loss of GA_1_). At 28 DPA, the loss of *t*Z and high levels of GA (GA_1_ and GA_4_) in olive fruit boost fruit growth mainly in terms of cell expansion, concomitantly with a rise in the level of IAA. Thus, IAA, GA_4_ and ABA have higher relative importance in the phase when olive fruit growth depends mainly on cell expansion, while CKs (*t*Z) and SA were greater during the phase of more intense cell division.

The role of auxins in early fruit growth has been amply reported [46,49,84]. In tomato, auxin signaling appears to be a prerequisite for the expansion of the fruit locular cells [47], for the negative control of the division of the fruit cells [85], and for fruit set [49,86]. A divergence was noted in auxin-related gene expression for cell division versus cell expansion during early olive fruit development (Figure 8; Appendix A). Transcripts involved in IAA synthesis, such as a transcript encoding tryptophan aminotransferase protein 4 (TAR4), were upregulated in the dividing olive fruit at 14 DPA (P14 vs. P0), while *YUCCA10* was downregulated during early olive fruit development (Figure 8). In addition, the IAA-amino acid hydrolase genes (*ILR1-like 1*, *ILR1-like 3*, and *ILR1-like 5*), involved in auxin homeostasis are exclusively overexpressed in the dividing olive fruit at 14 DPA (Figure 8). Auxin-amino acid hydrolase may provide local concentrations of auxin within the olive fruit during early development. However, expression levels for the auxin-conjugating enzymes *GH3.6*, *GH3.9*, and *GH3.10* were also found to be boosted in the dividing fruit (P14 vs. P0) during early fruit development. Furthermore, these *GH3.6* and *GH3.9* genes show downregulated expression in the expanding fruit (P28 vs. P14) during early olive fruit development, consistently with the highest IAA levels detected in the fruit at this time (Figure 8; Appendix A). In addition, this is consistent with the accepted idea that the release of auxins produced by seeds and/or surrounding fruit tissues triggers fruit growth through cell expansion [29,47]. Nevertheless, the *GH3.1* gene shows downregulated and upregulated expressions in the dividing fruit (P14 vs. P0) and in the expanding fruit (P28 vs. P14), respectively, during early olive fruit development (Figure 8; Appendix A). Similarly, during early olive fruit development, genes specifying auxin-signaling components (*TIR1*, *IAA9, ARF4*, *ARF8*, *ARF9*, *ARF18*, and *SAUR50)* were upregulated in both comparisons, while some genes (*IAA8, IAA26, ARF1*, *ARF2*, *ARF3*, *ARF5*, *ARF6 ARF10*, *ARF17*, *ARF19*, and *ARF22)* were exclusively upregulated in the dividing olive fruit (P14 vs. P0), and others (*IAA29* and *SAUR72)* were exclusively upregulated in the expanding olive fruit (P28 vs. P14) during early olive fruit development. Conversely, numerous auxin-regulated genes that were downregulated during early fruit development (*IAA14*, *IAA26*, *SAUR32*, *SAUR36*, *SAUR41*, *SAUR62*, *SAUR66, ARGOS*, *DAM1*, *DAM4*, and *PCNT115*, among others) were detected (Figure 8; Appendix A). Furthermore, transcript levels for two auxin efflux carriers, *PIN1* and *PIN7,* rose in the dividing fruit and lowered in the expanding fruit, while *PIL2*, *PIL6*, and *PIL7* expressions showed the opposite expression, suggesting that the latter participates in the transition from cell division to expansion during early olive fruit development. However, transcripts encoding auxin influx carrier-like proteins (*LAX2*, *LAX4* and *LAX5*) increased exclusively in the dividing fruit during early development, indicating altered auxin distribution in the fruit at this time. Therefore, the regulation of auxin conjugation, transport, and signaling may differ during early olive fruit development. Rises in endogenous IAA levels in the expanding olive fruit coincided with lower expression levels for the auxin conjugating enzymes, GH3.6 and GH3.9, and for two auxin efflux carriers, PIN1 and PIN7, while the transcript of IAA29, involved in auxin signaling, was upregulated exclusively in the expanding olive fruit, suggesting that *IAA29* may be a key candidate gene of auxin signaling during the cell-expansion phase in early-developing olive fruit.

GA, as well as auxins, reportedly serve as early signals for fruit set as well as for strong fruit growth [87]. A fainter GA stimulation, as well as weakened auxin signaling, has been reported to induce the change from a regular cell cycle to an endocycle [74]. In the case of GA, the present data suggest that GA synthesis is upregulated by *GGPS*, coding geranylgeranyl pyrophosphate synthase (GGPS), and by *GA20ox*, coding GA 20-oxidase; these two genes are active during early olive fruit development. Interestingly, *GA3ox* (GA 3-oxidase), which is a transcript involved in a late GA biosynthetic step, was downregulated in the dividing fruit (P14 vs. P0) and upregulated in the expanding fruit (P28 vs. P14), consistently with the highest GA (GA_1_ + GA_4_) levels detected in olive fruit at 28 DPA (Figure 8; Appendix A). Thus, these results suggest that GA3ox is the major late-step enzyme responsible for the high-level accumulation of GA_4_ detected in the expanding olive fruit, whereas the complete absence of GA_1_ observed in the dividing olive fruit may be attributed to the expression of the *GA2ox* gene, which is a transcript involved in the deactivation of bioactive GAs. Additionally, the data indicate that GA signaling is negatively regulated by *GAI1* and *RGL1* (DELLA proteins) in the cell-division phase, whereas GA signaling (*GID2* receptor) is upregulated in the expansion phase during early olive fruit development (Figure 8; Appendix A). 

These results agree with the established notion that GA participates mainly in the cell expansion process in the fruit [34]. Moreover, PACLOBUTRAZOL RESISTANCES 6 (PRE6), belonging to the bHLH TF family, which is involved not only in auxin, GA, and BR signaling, but also in light responses regulating cell expansion in Arabidopsis [88,89,90], was downregulated during early olive fruit development (Appendix A). The downregulated *PRE6* expression is hypothesized to promote cell expansion through GA-, auxin-, and BR-modulating response in olive fruit, as in Arabidopsis. In tomato, SlPRE2 and AtPRE1 proteins are involved in cell elongation through GA modulating response [91]. In this light, it is proposed that an increase in auxin and GA contents as well as GA signaling stimulation trigger the shift from a regular cell cycle to the endocycle in olive fruit, whereas the two hormones, auxin and CK, could activate the cell cycle in the developing fruit, according to absolute levels, cell sensitivity, and other signaling cross-talk. In fact, free CK hydrolyzed from its conjugates by CK riboside 5’-monophosphate phosphoribohydrolases (LOG) exclusively in the dividing fruit could in turn activate and coordinate the expression of cell division-related proteins (via ARR12). In particular, it bares noting that the upregulation of the expression of the three different *LOG3*, *LOG8,* and *LOG10* genes was found during the first 14 days of olive fruit development (P14 vs. P0), which is consistent with the highest *t*Z levels detected in olive fruit at 14 DPA (Figure 8; Appendix A). 

By contrast, both *CYP735A* (CK hydroxylase), which catalyzes the biosynthesis of trans-zeatin (*t*Z), and *ZOX*, which encodes zeatin O-xylosyltransferase involved in CK conjugation, were exclusively downregulated in the dividing olive fruit (P14 vs. P0) (Figure 8; Appendix A). However, transcript encoding for another zeatin conjugating enzyme *ZOG* (zeatin O-glucosyltransferase) increased exclusively in the expanding olive fruit at 28 DPA, consistently with the loss of *t*Z detected in the olive fruit at 28 DPA (Figure 7). These results suggest local transcriptional control of the CK biosynthesis and conjugation rather than transport from other tissues. Bioactive forms of CKs are, in turn, deactivated by CK oxidase/dehydrogenase (CKX), and thus regulate the amount of bioactive CK. In the present study, *CKX1* and *CKX7* gene expression were upregulated in the P14 versus P0 comparison, and downregulated in the P28 versus P14 comparison, while *CKX3* and *CKX9* gene expression were upregulated in both comparisons, suggesting that the deactivation of CK may actively occur during early olive fruit development (Figure 8; Appendix A). Therefore, the upregulated expressions of the three different *LOG3*, *LOG8,* and *LOG10* genes during the cell-division phase are probably sufficient to maintain *t*Z at levels higher than those that stimulated the fruit growth in olive, suggesting a key role of the LOG-dependent pathway in CK activation during the cell-division phase. Similarly, the upregulation of the expression of CK-responsive *ARR12* TF exclusively in the dividing olive fruit suggests that the *ARR12* gene activates CK-responsive gene expression during the fruit growth phase to ensure a prevalence of cell division over cell expansion. 

Furthermore, the BR signaling was clearly stimulated throughout early olive fruit development, since three *BRI1* receptor kinase, two *BAK1 (SERK2)* co-receptors, and three *BES1/BZR1* factor transcriptions were exclusively upregulated in the dividing olive fruit (P14 vs. P0), while only one different receptor *BRI1* gene was exclusively upregulated in the expanding fruit (P28 vs. P14; Figure 8; Appendix A), suggesting that the upregulation of receptor *BRI1* may be required for olive fruit development.

In the case of PA, the data imply that early olive fruit growth is mediated by upregulated PA biosynthesis and metabolism, including *ADC* (arginine decarboxylase), *AOase* (acetylornithine deacetylase), *OTCase* (ornithine carbamoyltransferase), *ACAULIS5* (thermospermine synthase), *SAMDC* (S-adenosylmethionine decarboxylase), *SPDS* (spermidine synthase), *SHT* (spermidine hydroxycinnamoyl transferase), *PUT* (polyamine uptake transporter), and *PAO* (polyamine oxidase) genes (Figure 8; Appendix A). This agrees with a prior study of the PA levels as well as the induction and spatial expression patterns of *SPDS* and *SAMDC* genes involved in spermidine biosynthesis during early olive fruit development [27]. In fact, both comparisons are apparently characterized by an active PA biosynthesis and conjugation (Figure 8; Appendix A), although the upregulation of two different PA transporter *PUT* (*PUT2* At1g31830 and *PUT4* At3g13620) genes, as well as two different *PAO* (*PAO2* and *PAO5*) genes exclusively in the dividing olive fruit, suggest that PA transport and catabolism may significantly influence the role of PA in the cell-division phase. Recently, the importance of PA transporters has been shown in the regulation of flowering and senescence pathways [92]. Substantial information is available on the PA transport systems in bacteria, yeast, and mammals, but the role of PA transporters in plants is only now being recognized [92]. 

By contrast, a downregulation of genes involved in ethylene (ET) and ABA biosynthesis occurred during early olive fruit development (Figure 8; Appendix A). Indeed, five S-adenosylmethionine synthase (*SAM1, SAM2, SAM3, SAM5*) genes, one 1-aminocyclopropane-1 carboxylic acid (ACC) synthase (*ACS*) gene, seven ACC oxidase (*ACO*) genes, and one ethylene-overproduction protein 1 (*ETO1*) gene were downregulated in the dividing olive fruit (P14 vs. P0), while two *SAMS* (*SAM1*, and *SAMS2*) genes, one ACC synthase (*ACS2*) gene, and seven ACC oxidase (*ACO*) genes were downregulated in the expanding olive fruit (P28 vs. P14), suggesting that the early olive fruit development can apparently be characterized by an inactive ET biosynthesis at the transcriptional level. Similarly, other genes related to ET signaling, such as *EIN4, EIN2, CTR1, EIL3* (EIN3*-like*, *EIL*), and *ERF* (*ERF001*, *ERF004*, *ERF005*, *ERF010*, *ERF011*, *ERF027*, *ERF104*, *ERF110*, *ERF113*, *ERF116*, *ERF118*), among others, were also downregulated during early olive fruit development (Figure 8; Appendix A). Likewise, the upregulation of the genes related to ET signaling during early olive fruit development was also found (Figure 8; Appendix A): ERF011 and ERF038 were related to early olive fruit growth, whereas ERF12, ERF107, and ERF118 were likely to be associated with a period of more intense cell division, and ERF027, ERF061, and ERF071 tended to be associated with the cell-expansion period. 

Interestingly, the present analysis also found components of ABA signaling repressed during early fruit development, such as two *PYR/PYL (PYL8* and *PYL4,* ABA receptors) genes and one *ABI5 (*bZIP TF) gene. By contrast, one *PYR/PYL (PYL9)* and seven *PP2C* genes were exclusively upregulated in the dividing olive fruit (Figure 8; Appendix A), suggesting that this PYL9 receptor forms part of an ABA signaling unit that modulates early olive fruit growth. Thus, these results suggest that ABA signaling was downregulated during the cell-expansion phase, and one *ABI5* gene is the major TF controlling ABA-mediated gene expression during olive fruit growth.

Likewise, several genes related to SA were upregulated exclusively during the first 14 days of fruit development, such as *PAL*, *NPR2*, and *PR-1* genes, consistently with the highest SA levels detected in the fruit at 14 DPA. Meanwhile, other genes such as *SAMT, NPR1, NPR3*, and *NPR5* genes were downregulated in the dividing olive fruit (P14 vs. P0) (Figure 8; Appendix A), but one *SAMT* transcript, which encodes increased SA carboxyl methyltransferase (an enzyme responsible for biosynthesis of methyl salicylate) in the expanding olive fruit, was in good agreement with a loss of SA in the expanding olive fruit at 28 DPA (Figure 7; Appendix A). Conversely, *JMT* genes encoding a jasmonate-O-methyltransferase, which catalyzes the methylation of JA into methyljasmonate (MeJA), were also upregulated during early fruit development, suggesting that the MeJA could regulate the cell growth rate during early olive fruit development. JA conjugation with isoleucine is reportedly catalyzed by jasmonoyl-isoleucine (JA-Ile) synthetase (JAR1), which is a member of the GH3 family [93]. The most bioactive of the Jas is JA-Ile. The perception of JA-Ile by its coreceptor, the Skp1-Cullin1-F-box-type (SCF) protein ubiquitin ligase complex SCF^COI1^-JAZ, derepresses the transcriptional repression of target genes in the nucleus. JA-Ile signaling participates in regulating several developmental processes, such as root growth and architecture, tuber and trichome formation, seed germination, and particularly reproductive-organ development [94]. Here, it is reported that *JAR1*, *COI1, NINJA*, and *TPL* (TOPLESS) genes were downregulated in the olive fruit during early development (Figure 8; Appendix A), suggesting that these components may aid in JA-Ile signaling in the nucleus to regulate early fruit development. Notably, these results also suggest that *AOS1*, which encodes an allene oxide synthase (AOS) protein, is the major control point responsible for the drop in JA detected in olive fruit during early development. AOS, the second enzyme in the biosynthesis of the JA, is a regulatory point in the biosynthesis of JA [95]. Interestingly, this analysis showed that *LOX3* (13-LOX member) and *LOX5* (9-LOX member) genes encoding lipoxygenase members were downregulated in the olive fruit during early development, whereas *LOX2* (13-LOX member) and *LOX6* (13-LOX member) were exclusively upregulated in the expanding olive fruit at 28 DPA (Figure 8; Appendix A), indicating that these genes modulate the distribution between 13-LOX- and 9-LOX-derived oxylipins during early fruit development.

Finally, the identification of the proteins involved in oxide nitric (NO) formation/transport and strigolactone (SL) signaling in olive fruit constitutes a promising avenue of research for a better comprehension of NO and SL physiological functions during early fruit development. Here, several transcripts related to NO metabolism were also differentially expressed during early olive fruit development, such as *NOS* genes encoding putative NO synthase, and *NR2* genes encoding putative nitrate reductase (NR), which were upregulated and downregulated during early fruit development, respectively (Figure 8; Appendix A). Similarly, *NRT*2.5 and *NRT2.7* gene-encoding member nitrate transporters were downregulated in the olive fruit during early development. In addition, genes involved in the SL signaling pathway, such as the receptor *DAD2* (SL esterase), were upregulated exclusively in the dividing olive fruit at 14 DPA (P14 vs. P0), whereas the receptor *D14* (α/β hydrolase DWARF14) and the transcriptional repressor *SMXL* (SUPPRESSOR OF MAX2 LIKE) genes were exclusively downregulated in the dividing olive fruit during early development, suggesting that these components act to regulate karrikins/strigolactone responses during the cell division phase of early olive fruit development. In contrast to this, it is shown here that one *MAX2* gene encoding MORE AXILLARY GROWTH2 (MAX2), an F-box protein that is part of a SKP1-Cullin-F-box (SCF) ubiquitin ligase complex, was exclusively downregulated in the expanding olive fruit at 28 DPA (Figure 8; Appendix A). 

Therefore, the comprehensive study of varying gene expression, together with analyses of different hormonal composition, unveils complex hormone control underlying early fruit growth in olive, with intricate temporal variations, implying distinct regulatory programs. Key candidate genes involved in CK (*LOG3*, *LOG8*, *LOG10, CKX1, CKX7,* and *ARR12*) and auxin (*LAX2*, *LAX4*, *LAX5, PIN1, PIN7, GH3.6, GH3.9, GH3.10, IAA8, IAA26, ARF1, ARF2, ARF3*, *ARF5*, *ARF6*, *ARF10*, *ARF17*, *ARF19*, and *ARF22)* metabolism and signaling showed a preferential expression in the cell-division phase, while some other candidate genes encoding proteins with roles in auxin (*PIL2*, *PIL6*, *PIL7, IAA29*, *GH3.1,* and *SAUR72)* and GA (*GA3ox* and *GID2*) metabolism and signaling could act in the cell-expansion phase during early olive fruit growth, suggesting that CKs as well as auxins positively affect cell division, whereas the GA and auxin hormones display positive effects on cell expansion during early fruit development in olive. Likewise, cross-talk between CKs, GAs, and auxins stimulate cell growth that is likely in the developing olive fruit where co-expression of the following occurs: *YUCCA10* (auxin synthesis); *TIR1*, *IAA9*, *ARF4, ARF8*, *ARF9, ARF18,* and *SAUR50* (auxin signaling); *PIN1* and *PIN5* (auxin transport); *GGPS* (GA synthesis); *CKX3* and *CKX9* (CK metabolism); *HK3*, *HK4,* and *ARR5* (CK signaling). In addition, the data imply that both cell-division and cell-expansion phases are mediated by upregulated BR signaling (receptor *BRI1)*, and downregulated ABA (*ABI5*) and ET (*ERF001*, *ERF004*, *ERF005*, *ERF010*, *ERF011*, *ERF027*, *ERF104*, *ERF110*, *ERF113*, *ERF116*, and *ERF118)* signaling, as well as downregulated JA biosynthesis (*AOS*). In contrast, the cell division is mediated primarily by upregulated SA synthesis (*PAL*), auxin (*PIN1* and *PIN7*) and PA (*PUT2* and *PUT4)* transport, GA catabolism (*GA2ox*), and CK signaling (*ARR12*), and the cell expansion is mediated mainly by upregulated GA biosynthesis (*GA3ox*), as well as upregulated auxin (*IAA29*) and GA (*GID2*) signaling during early olive fruit development.

### 2.7. Signaling Peptides Regulating Early Fruit Development in Olive

The molecular identifications for the signaling peptides during early fruit development are still limited. Signaling peptides in the role of phytohormones control several features of plant growth and development through cell–cell communication networks, e.g., organ abscission, meristem maintenance, gravitropism, cell proliferation and differentiation, cell elongation, and defense [96]. These peptides can be recognized typically by membrane-embedded receptor-like kinases which activate cell signaling to govern plant growth and development. In the present analysis, more than 40 genes encoding putative leucine-rich repeat receptor-like kinases (LRR-RLKs) family proteins, involved in plant peptide signaling, differed in expression level during early olive fruit development (Appendix A). Among these, *EMS1* gene encoding for the LRR-RLKs family protein was found to be exclusively detected in the dividing olive fruit. Similarly, one cysteine-rich receptor-like protein kinase 29 (CRK29) and one probable receptor-like protein kinase At5g38990 (MDS1), which are involved in growth adaptation upon exposure to metal ions [97], were found to be detected exclusively in the dividing olive fruit (Appendix A). Likewise, more than 400 genes have been found to encode putative proteases (also referred to as peptidases or proteinases), including serine proteases, cysteine proteases, aspartic proteases, and Clp proteases, among others (Appendix A). Proteases are enzymes able to participate in almost all stages of plant life [96,98]. In particular, two genes encoding for serine carboxypeptidase-like 45, and serine carboxypeptidase-like 2 were found to be detected only in the dividing olive fruit (Appendix A).

Moreover, two genes, *SBT5.4*, and *SBT4.15*, coding for subtilisin-like protease proteins (also known as subtilases; SBT) were found to be detected exclusively in the flower at anthesis (fruit set), and in the dividing fruit at 14 DPA, respectively (Appendix A). Although previous studies have indicated that SBT-mediated processing results in the activation of peptide signals regulate stress-induced flower drop, the formation of the embryonic cuticle, and pollen development [98], little has been reported concerning its role in early fleshy fruit development. 

The present analysis identified 32 putative SBT genes in olive fruit that differed in expression level during early fruit development (Figure 9A,B), and were retained for further phylogenetic analysis (Figure 9C). Among 32 SBT genes identified in the analysis, 11 were exclusively expressed in the dividing olive fruit at 14 DPA, while 4 were expressed only at 28 DPA (Figure 9; Appendix A), indicating that at least some members of SBTs play major roles during early olive fruit development. Overall, 19 SBT genes were upregulated and 9 were downregulated in fruit at 14 DPA compared with 0 DPA, whereas 11 SBT genes were upregulated and 8 were downregulated in the expanding fruit at 28 DPA compared with 14 DPA (Figure 9; Appendix A), of which five genes were randomly examined and their expression patterns were confirmed using qRT-PCR (Figure 9D). Notably, it is shown that, after fertilization, transcripts that encode SBT1.1, SBT1.3, SBT1.5, SBT1.6, and SBT1.8 are associated with both cell division and cell expansion phases, whereas the cell division phase involves mainly the gene activity of SBT1.2, SBT1.9, and SBT3.5 proteins. Furthermore, the cell expansion phase is associated with the gene activity of SBT2.5 and SBT3.6 proteins during early olive fruit growth (Figure 9; Appendix A). Previously, SBT3.5 and SBT6.1 proteins have been shown to promote cell expansion [99], but cross-talk involving different signaling peptide pathways controlling cell expansion has not been elucidated, nor has their relation to hormonal growth control.

### 2.8. Transcript Changes in Cell Wall Biosynthesis and Remodeling during Early Olive Fruit Development

In this study, among the DEGs, 895 genes were identified that encode proteins with probable functions in cell wall biosynthesis and remodeling during early olive fruit development (Appendix A), of which 300 were upregulated and 258 were downregulated in the P14 versus P0 comparison, and 195 were upregulated and 142 were downregulated in the P28 versus P14 comparison (Appendix A). Overall, 119 genes were upregulated, and 75 genes were downregulated in both comparisons during early fruit development (Figure 10; Appendix A). The well-represented families included cellulose synthase (CES, 59 genes), arabinogalactan protein (AGP, 59 genes), polygalacturonase (PG, 55 genes), pectin methylesterase (PME) (53 genes), glucan 1,3-β-glucosidase (BG; 43 genes), endo-1,4-*β*-glucanase or cellulase (EGase/CEL, 41 genes), expansin (EXP, 39 genes), xyloglucan endotransglucosylase/hydrolase (XTH, 37 genes), laccase (LAC, 34 genes), β-galactosidase (βGAL, 28 genes), and extensin (EXT, 28 genes) proteins (Figure 10; Appendix A), implying that these cell wall-related enzymes help regulate early olive fruit development.

EGase, XTH, and EXP are cell wall-loosening factors expressed during the stages of maximum growth in fleshy fruit [46,50,53]. An examination of these families identified genes that were upregulated exclusively in the dividing olive fruit (14 DPA), such as genes that encode one EXP (EXPA13), four EGase/CELs, two XTHs (XTH9, XTH10), five CSs (CESA2, CESA3, CESA6, CESE6, CESG2), two AGPs (AGP26, AGP31), three EXTs (EXT2, EXT3, EXT6), and two LAC (LAC15, LAC7) proteins (Figure 10; Appendix A). Similarly, genes upregulated only in the expanding olive fruit (28 DPA) were identified, such as genes encoding two EXP (EXPA15, EXPA8), two EGases/CEL (CEL10, CEL25), two XTH (XTH32, XTH33), one CS (CESD4), four AGP (AGP10, AGP19, AGP20, AGP26), and one EXT (EXT1) proteins (Figure 10; Appendix A), suggesting that these cell wall-related enzymes regulate cell expansion during early fruit development in the olive.

Therefore, the analysis reveals a sequential induction of cell wall biosynthesis and remodeling genes during early olive fruit development, as well as the distinctiveness of candidate genes associated with cell division in comparison with cell expansion during early fruit growth. 

### 2.9. Characterization of Transport-Related Genes Associated with Early Olive Fruit Development

Furthermore, during early fruit development in olive, significant change was identified in the abundance of a subset of 1187 gene-encoding channel and transporter proteins of 127 diverse families (Appendix A). Likewise, of the 127 different families of channel and transporter proteins, 26 were expressed only in the dividing olive fruit including the dicarboxylate transporter, GDP-fucose transporter, membrane magnesium transporter, and nucleobase-ascorbate transporter families, 25 families (the protein EXORDIUM, protein YLS3-like, protein NRT1/PTR FAMILY, lipid-transfer protein, sulphite exporter, tetraspanin-2, glycerol-3-phosphate transporter, and glycolipid transfer protein families, among others) are regulated only in the expanding olive fruit, and 76 families (ATP-binding cassette (ABC) transporter, amino acid transporter, aquaporin, glucose-6-phosphate/phosphate translocator, hexose carrier protein, sugar transporter, boron transporter, CMP-sialic acid transporter, copper transport protein, GABA transporter, and magnesium transporter families, among others) are regulated in both developmental stages of olive fruit (Figure 11; Appendix A).

Among the most abundant upregulated genes in the dividing olive fruit (P14 vs. P0), a homologue of *KT12* (AT1G60160 putative orthologue) encoding a member of the KT/KUP/HAK family of proton-coupled potassium transporters (KT) was identified, these being highly expressed also in the expanding olive fruit (28 DPA). KT12 have potential effects on cell expansion regulated by auxin [100]. Overall, 26 genes that encode KTs fruit (Figure 11; Appendix A) were identified. Of these, two genes (*KT5*, *KT8*) were upregulated exclusively in the P14 versus P0 comparison, one gene (*KT2*) was exclusively upregulated in the P28 versus P14 comparison, and two genes (*KT6*, *KT12*) were upregulated in both comparisons. Vigorous cell expansion occurs with greater cell-turgor pressure, which depends on the controlled import of ions to promote water influx. Given that, in plant cells, potassium is the predominant inorganic ion, K transporters and channels constitute the most plausible candidates involved in the production of osmotic gradients through the plasma membrane and thus the molecular mediators responsible for vigorous cell expansion [101]. However, only limited research is available concerning which channels/transporters are involved in K import during early fruit development. 

Among the 162 genes related to transport and upregulation in the expanding olive fruit at 28 DPA, the most abundant upregulated genes encode one nonspecific lipid-transfer protein (nsLTP2) (Figure 11; Appendix A), which is found in seeds and involved in the transport of the more rigid suberin monomers and sterols [102]. This result may mean that this OeLTP2 protein facilitates the rapid transport of lipids to the plasma membrane in the fruit during early growth, which may then remain at the cell surface to defend against pathogens. 

Furthermore, in the present study, three sphingolipid transporter genes encoding SPNS2 proteins were exclusively upregulated in the dividing fruit (Figure 11; Appendix A), in agreement with a prior study concerning the sphingolipid levels during early olive fruit development [62], whereas several phospholipid transporting ATPase genes were upregulated in the dividing fruit at 14 DPA, and downregulated at 28 DPA (Figure 11; Appendix A), indicating that these lipophilic membrane components are essential for diverse cellular functions during early olive fruit growth. Thus, the metabolism during early fruit development is highly active in this species, including alterations in membrane lipid metabolisms and transport.

### 2.10. Identifying Transcription Factors Critical for Early Olive Fruit Development

Identifying new TF genes as well as their function in regulating the expression of candidate genes will aid a fuller understanding of the signaling pathways regulating early olive fruit development. Of the DEGs studied, 1150 genes presumably encoding TFs of diverse families were differentially expressed during early fruit development in olive, most of which had a downregulation pattern from 0 to 28 DPA (Figure 12A; Appendix A). In particular, 902 DEGs were within the P14 vs. P0 comparison (the set of cell division-related genes), and 564 genes were within the P28 vs. P14 comparison (the set of cell expansion-related genes). The RNA-seq analysis revealed that 385 and 517 genes were upregulated and downregulated in the dividing olive fruit (the P14 vs. P0 comparison), respectively, while 174 and 390 genes were upregulated and downregulated in the expanding fruit (the P28 vs. P14 comparison), respectively (Appendix A). These DEGs were shown to be especially related to the TFs MYB, bHLH, ZF, homeobox domain proteins, WRKY, and bZIP families (Figure 12A), implying that TFs from these families may take part in triggering the transcriptional cascade during early fruit development in olive.

The set of cell division-related genes is especially enriched in ZF proteins, whereas set of cell expansion-related genes was found to be rich mainly in the MYB and bHLH families (Figure 12A). Hence, despite the fact that two sets contained members from a number of TF families, clearly significant differences were found in each set regarding the proportion of families (Figure 12A; Appendix A). In addition, distinct TF families make up each set: the Whirly, Trihelix, and SNF2 (Helicase-like TF or CHromatin Remodeling Protein) families in the set of cell division-related genes, and the ARID/BRIGHT and VOZ families in the set of cell expansion-related genes (Figure 12A; Appendix A). Thus, the results indicate that the cell division phase is potentially controlled by the upregulation of E2F/DP, Whirly, TCP, and homeobox TFs, while the cell expansion phase may be controlled by the upregulation of MADS-box and GATA-binding proteins during early fruit development (Figure 12A; Appendix A). However, most members of the zinc finger (ZF), the basic lecine zipper (bZIP), GRAS, WRKY, NAC, PLATZ, and AP2/ERF families were downregulated genes during early fruit development in olive. In different gene groups from each set, the enrichment of sequence elements together with data concerning transcript abundance provide a feasible set of TFs that are able to bind these elements and that may be valuable as a focus of future research.

Within the set of cell division-related genes (P14 vs. P0 comparison), three TFs (TFs) homologous to *AINTEGUMENTA (ANT)*, *bHLH61*, and *MYB3R-1* belonging to the AP2/ERF, bHLH, and MYB families, respectively, have been identified as the most abundant TFs. Indeed, these TF families were among the most widely represented class of proteins in the dividing olive fruit (Figure 12A; Appendix A). In addition, three bHLH (bHLH18, bHLH87, and bHLH93), and one ZF CCCH domain-containing (C3H2) protein were the most abundant TFs found in the expanding fruit (the P28 vs. P14 comparison). Of particular interest is ANT, one member of the AP2 family, which was the most abundant TF transcript expressed in the dividing olive fruit (Appendix A). ANT is required in the control of cell proliferation in Arabidopsis, and regulates growth and cell numbers during organogenesis, modulating auxin biosynthesis in the ovule via regulation of YUCCA4 [103]. In the present work, on the base of homology and the strong upregulation of *ANT* exclusively in the dividing olive fruit, it is hypothesized that this TF plays a major role in the regulation of cell division and size through its downstream control of auxin biosynthesis in the fruit, consistently with the IAA levels detected in the dividing fruit at 14 DPA (Figure 7). Previously, it has been suggested that *CYCD3;1* is a target of ANT [104]. *CYCD3* genes regulate the length of the temporal period of mitotic cell division during aerial organ development, and are induced by CKs in Arabidopsis [105]. According to the data, the *CYCD3;1*, *ANT*, and CK metabolism, together with signaling gene expression, share a common shift during early olive fruit growth (Figure 8), indicating that these components may act in conserved pathways linking CKs to developmentally regulated cell division in olive fruit enriched mainly in CKs (*t*Z). 

In Arabidopsis, researchers have identified a small number of organ-size regulators. Notably, among the specific genes in the dividing olive fruit, two *Growth-Regulating Factor* (GRF) genes, GRF1 and GRF3, which encode a transcription activator that plays a role in the regulation of cell proliferation and size in Arabidopsis leaves [106], abundantly accumulated in the dividing olive fruit at 14 DPA (Appendix A), indicating the potential roles of these genes in signaling pathways for the regulation of early olive fruit growth. A previous study has shown GRF3 to be a component in a network composed of miR396, GRFs, and their interacting factors (GIFs) contributing to at least partial regulation of meristem function by controlling cell proliferation after infection by cyst nematode [107]. In tomato plants expressing higher levels of *SlGRF1* to *-5*, fruit size and weight are increased as a result of an increased size of the epidermal cells [108]. The present findings provide a valuable tool for unravelling the physiological function of GRF1 and GRF3 in future studies. Conversely, negative effectors of cell proliferation in Arabidopsis, such as BIG BROTHER (BB) [109], was also identified in the present analysis. Here, eight *BB* genes were upregulated in the expanding olive fruit at 28 DPA (Appendix A), indicating that at least some members of BBs play a role not only in limiting the cell-proliferation stage, but also in fostering the transition towards cell differentiation and expansion in early olive fruit growth. Nevertheless, the way in which these different effectors act on the regulatory genes of the cell cycle in developing olive fruit has not been elucidated.

Furthermore, the major upregulated differentially expressed TFs (Log_2_ Fold Change > 3) were identified during early olive fruit development. As shown in Figure 12B, 26 TFs that might be associated with cell division phase were identified, including 2 DOF-ZF (DOF3.4 and DOF4.7), 2 MYB (MY8 and MYB20), 5 WRKY (WRKY6, WRKY14, WRKY28, WRKY35, and WRKY71), 2 MADS-box (SOC1 and AGL42), 2 NAC (NAC29), and 1 AP2/ERF (ERF110) proteins, among others, while 16 TFs might be involved in cell expansion phase, including 2 bHLH (bHLH87 and bHLH162), E2FE, WRKY13, NAI1, 2 MYB (MYB14 and MYB83), 2 AP2/ERF (ERF071 and ERF027) proteins, among others.

According to their expression profiles, other genes appear to be promising candidates for the main regulations in the early development of olive fruit. In the RNA-seq analysis, four TFs, SCREAM2 (or bHLH33), bHLH18, bHLH71, and Zm1 (R2R3-MYB), were identified within the differentially expressed gene set, and were all upregulated in both the dividing and the expanding olive fruits (P14 vs. P0, and P28 vs. P14 comparisons) and were thus linkable to cell division and cell expansion phases during early fruit development in olive (Figure 12B; Appendix A). SCREAM2 (or bHLH33, a putative orthologue At1g12860) encoded a key component of core regulatory units in stomatal development, most likely by controlling successive roles of SPCH, MUTE, and FAMA (bHLH proteins) in Arabidopsis [110]. Additionally, SCREAM is INDUCER of CBF EXPRESSION1 (ICE1), a bHLH TF that positively regulates the cold-induced transcriptome and freezing tolerance [110]. According to the analysis, a homologue of *ICE1* is also upregulated in both comparisons, suggesting that the two proteins prompt sequential steps in the development program (i.e., entry, proliferation, and terminal differentiation) of the stomatal cell during early fruit development in olive. Moreover, here it was found that other members of the bHLH TF family, such as bHLH61 (At5g10570), are exclusively upregulated in the dividing olive fruit, and bHLH87 (At3g21330) and bHLH162 (At4g20970) are exclusively upregulated in the expanding olive fruit. The bHLH proteins represent prime regulatory components in networks of transcription that control several biological processes, such as fruit and flower development [111,112,113]. Therefore, changes in the temporal distribution of these proteins could lead to unexpected cross-talk in the regulatory network of genes.

Moreover, bHLHs and MYBs or bZIPs bind to DNA as either homodimers or heterodimers, multiplying the feasible regulatory combinations [114]. In this context, MYB and bZIP TFs are members of TF families abundantly represented in the developing olive fruit (Appendix A). In the dividing olive fruit, the results demonstrated upregulation of 33 (MYB20, MYBS3, among others) out of 122 MYB genes identified, and 17 (bZIP11, bZIP44, among others) out of 49 bZIP genes identified, while 20 (MYB14, MYB83, among others) MYB genes and 4 bZIP (bZIP6, bZIP11, bZIP18, and RF2b) genes were upregulated in the expanding olive fruit. The possibility that bHLH proteins act as an interaction partner for MYB or bZIP TFs in regulating genes involved in processes downstream during early olive fruit growth should not be disregarded. More research is needed in order to determine whether or not these bHLH TFs participate jointly with MYB or bZIP proteins in the signaling of developing olive fruit. From this perspective, MYB TFs appear to take part in controlling the cell cycle not only in animals, but also in plants and other higher eukaryotes [115]. MYB factors are involved in regulating the transcription of cyclin genes via MYB recognition elements in cyclin promoters [116]. The RNA-Seq analysis showed that the upregulation of MYB3R-4 (3R-MYB, AT5G11510.1), which activates mitotic gene expression and CK response [117], and MYB3R-5 (3R-MYB, AT5G02320.1), which is a member of the DREAM complex [118], are probably associated with the cell division phase, while the upregulation of MYB3R-1 (3R-MYR, AT4G32730.2) is likely associated with the cell expansion phase during early olive fruit development (Appendix A). Additionally, members of the R2R3-MYB family have been identified in olive fruit, as in the case for Zm1, which is involved in the regulation of flavonoid biosynthesis [119]. The increased expression of *Zm1* in both the dividing and expanding olive fruits suggests the involvement of Zm1 in early fruit growth. Likewise, many R2R3-MYB genes also act in signal transduction pathways of ABA, GA, JA, and SA [115], supporting the contention that MYB proteins constitute vital components of many hormone-mediated transcriptional cascades, such as ABA, GA, and SA, which regulate early fruit development in olive. Thus, the present study reveals novel regulators associated with early fruit growth and provides a useful resource for further characterization of the physiological functions of these regulators in olive and other plant species.

## 3. Materials and Methods

### 3.1. Plant Material and Cytological Study

Olive trees (*Olea europaea* L.) of the ‘Picual’ cultivar 20 years old in an orchard under drip irrigation (fertigation with suitable liquid fertilizers) near Badajoz (Spain) were studied. From these trees, flowers were collected at the anthesis stage (0 days post-anthesis, DPA) and fruits were sampled at 7, 14, 21, 28, 35, and 42 DPA, spanning a time from the fruit set to the time of endocarp lignification, and 500 flowers or fruits for each developmental stage during the 2018–2019 growing seasons [27,62,63]. Harvested flowers and whole fruits of ‘Picual’ olive were weighed, and the longitudinal and transverse diameters were measured at different developmental stages (Figure 1). Flowers plus whole fruits at different stages of development were frozen in liquid nitrogen and then stored at −80 °C until analyzed.

In the cytological study, at least three biological replicates were made at each stage. For this, samples were sliced 0.3 to 0.6 mm thick and quickly fixed in ethanol-acetic acid (3:1, *v/v*) for 2 h at an ambient temperature. Then, the samples were rinsed 3 times using 70% ethanol, dried using an ethanol series, and finally embedded in Technovit 7100 (Kulzer). Sections that were 30 μm thick were used for confocal imaging and were stained for 2 min with Calcofluor White M2R (Wyeth, http://www.wyeth.com (accessed on 10 January 2020), 0.1% *w/v* in distilled water), a fluorescence brightener commonly used in the visualization of crystalline cellulose in plant cell walls [120], rinsed with distilled water, mounted with a cover slip in DABCO mounting medium, and sealed with nail polish. Fruit cells and pericarp thickness were examined using a confocal laser-scanning microscope (model Fluoview-BX50, Olympus). For microscopy, the cell size (cell area) and pericarp thickness (cross-section) were determined using a CellProfiler image analysis system [121].

### 3.2. Flow Cytometry Analysis

The cell division period was precisely determined using flow cytometry with nucleus ploidy profiles taken from ovaries and fruits using the method of Reference [122]. The samples of 0.1–0.2 g fresh weight were diced into 0.5 mL using a blade and placed in an ice-cold buffer (0.2 m Tris-HCl, 4 mm MgCl_2_, 2 mm EDTA, 86 mm NaCl, 10 mm metabisulphite, pH 7.5, and 1% Triton X-100) then filtered through a 30-µm nylon mesh, and finally stained using 4,6-diamidino-2-phenylindole (DAPI). Next, the distribution of the nuclear-DNA content was analyzed with a FACS CantoII flow cytometer, and the recorded data were analyzed using FACSDiva 6.1.2 software (BD Biosciences, Franklin Lakes, NJ, USA). For each stage, 4 biological replicates, with 10,000 nuclei each, were undertaken. The DNA content (C value) of the olive fruit cells was submitted to flow cytometry with the use of internal calibration standards [122]. The C values were determined following [123]: 2C DNA (pg) = (mean of the problem sample G1 peak × 2C DNA content of the standard [pg])/mean of the standard G1 peak. 

### 3.3. RNA Extraction, Library Preparation, and Sequencing

The extraction of the total RNA from ovaries and fruits samples was performed as in [27]. For each sample, three biological replicates were collected of the same developmental stage. Tissue frozen at −80 °C was milled into a powder in liquid nitrogen prior to the addition of the buffer (2 mL buffer: 1 g tissue). The RNA quality tests were performed with the Agilent 2100 Bioanalyzer (Agilent Technologies, Santa Clara, CA, USA) employing an RNA 6000 Pico assay kit (Agilent Technologies). This RNA was used in RNA-seq and quantitative RT-PCR (qRT-PCR) analyses. The poly(A)+ mRNA fraction was isolated from the total RNA and obtained cDNA libraries following the recommendations of Illumina (Truseq stranded Illumina kit). In short, poly(A)+ RNA, after being isolated on poly-T oligo attached magnetic beads, was chemically fragmented before reverse transcription and cDNA generation. Next, in the end-repair process of the cDNA fragments, a single ‘A’ base was added to the 3′ end, and the adapters were ligated afterwards. Finally, the products were purified and enriched with PCR to compile the final indexed double-stranded cDNA library. The quality of the libraries was checked using a Bioanalyzer 2100, High Sensitivity assay. The quantity within the libraries was assessed by real-time PCR in LightCycler 480 (Roche). Before cluster generation in cbot (Illumina), an equimolar pooling of the DNA libraries was performed and the pool was then sequenced using paired-end sequencing (100 × 2) with an Illumina HiSeq 2000 sequencer. The RNA-seq analysis was ordered from Sistemas Genómicos (Valencia, Spain). Three independent biological replicates (three olive trees) per sample were sequenced, each replicate consisting of an equilibrated pool of three RNAs from three different samples per replicate.

Quality control checks of raw sequencing data were performed with FastQC 0.11.7. The generated reads were cleaned using the Trimgalore 0.6.2 method (http://www.bioinformatics.babraham.ac.uk/projects/trimgalore/, accessed on 10 January 2020). Subsequently, all samples were combined, and the complexity of the reads was reduced by removing duplicates and low-quality reads using Picard tools 2.9.2 and Samtools 1.9 algorithms [124] (>Q20). Reads were mapped to the *Olea europaea* var. sylvestris genome as the reference [*Olea europaea* var. sylvestris Annotation Report (nih.gov, accessed on 10 January 2020)] [69]. Each sample was then mapped with the latest bowtie2 2.3.3 version [125]. The reads of good quality (Phred > 20) were selected to increase the resolution of the count expression. Finally, the expression inference was evaluated by means of the count of properly paired end reads by transcript.

### 3.4. Differential Expression

For the study of differential gene expression and normalization gene expression using library size, the DESeq 1.36 algorithm was used [126]. This method is based on a negative binomial distribution. The genes with a fold change of less than −2 or greater than 2 and a *p* adjust by FDR < 0.05 were considered differentially expressed [127]. A functional enrichment study was performed using the information obtained from the pFam and uniport databases. This study is based on a hypergeometric distribution using the statistical software R. The principal-component analysis (PCA) was analyzed using the methods described in DESeq using the normalized counts of gene expression obtained from this method. Differential expression was made for each comparison. Then, specific gene expression from each comparison was obtained using a venn diagram (https://bioinfogp.cnb.csic.es/tools/venny/, accessed on 10 January 2020). The GO enrichment was performance using a hypergeometric test based in R package. KEGG data was accessed using AnnotationHub [128,129]. 

The alignment of the deduced protein sequences and the phylogenetic tree were computed using the ClustalW [130] in MEGA11 software (http://www.megasoftware.net, accessed on 10 January 2022).

### 3.5. Quantitative RT-PCR

The RNA-seq results by qRT-PCR were validated as indicated in [18]. The primers employed are shown in Appendix A. Relative fold differences were calculated based on the comparative Ct method using the *Olea europaea Ubiquitin* (*OeUb)* as an internal standard [27]. At least two to three independent RNA isolations were used for cDNA synthesis, and each cDNA sample was subjected to real-time PCR analysis in triplicate. 

### 3.6. Quantification of Plant Hormones

A pool of 100-mg fresh weight/sample was used for each measurement split into 3 independent biological replicates per sample. The plant hormones were quantified as described by Reference [131]. Aliquots of lyophilized material were extracted with 80% methanol-1% acetic acid. Deuterium-labeled hormones (purchased from Prof. L Mander- Canberra, OlChemim Ltd-Olomouc, or Cambridge Isotope Lab- Andover) [17,17-^2^H]GAn, [^2^H_5_]IAA, and [^2^H_6_]ABA were added as internal standards for quantification of SA and ABA. For quantification of JA, the compound dhJA was used instead. For collecting the acid fractions containing SA, ABA, and JA, the extracts passed consecutively through HLB (reverse phase), MCX (cationic exchange), and WAX (ionic exchange) columns (Oasis 30 mg, Waters). The final residue was dissolved in 5% acetonitrile-1% acetic acid, and the hormones were separated by reverse phase UPHL chromatography (2.6 μm Accucore RP-MS column, 100 mm length × 2.1 mm i.d., ThermoFisher Scientific) with a 5% to 50% acetonitrile gradient. The hormones were analyzed by electrospray ionization and targeted-SIM using a Q-Exactive spectrometer (Orbitrap detector, ThermoFisher Scientific, Spain). The concentrations of hormones in the extracts were determined using embedded calibration curves and the Xcalibur 4.1 SP1 build 48 and TraceFinder programs. 

## 4. Conclusions

Cytological, ploidy, hormonal, and transcriptional analyses reveal the rate and duration of cell division and cell expansion phases during early olive fruit growth as well as the regulation of these phases. The present study provides evidence that CKs may regulate cell division mainly through the *ARR12* expression gene, and that CKs and SA appear to correlate positively with cell division during early olive fruit development, while GA (via *GA3ox* and *GID2)*, auxin (via *PIL2*, *PIL6*, *PIL7, IAA29*, *GH3.1,* and *SAUR72),* and ABA (via *ABI5*) have higher relative importance in the period where fruit growth depends mainly on cell expansion. In particular, CK (*t*Z) and GA_4_ present different accumulation patterns during early olive fruit development, suggesting that the *tZ* and GA_4_ hormones act on early olive fruit development in opposite ways during the transition from cell division to cell expansion. Furthermore, this study offers the first detailed analysis available for an array of cellular responses under the control of gene expression leading to early fruit development in olive. Through gene expression studies, a selection was made of the candidate genes having well-known molecular roles in different biological processes affecting olive fruit growth. This gene expression profile, together with hormonal regulators, will help clarify gene regulatory networks during early olive fruit development. In practical terms, it will aid in designing specific approaches for crop breeding and engineering, facilitating the development of the table olive and olive oil industries.

## Figures and Tables

**Figure 1 ijms-24-00961-f001:**
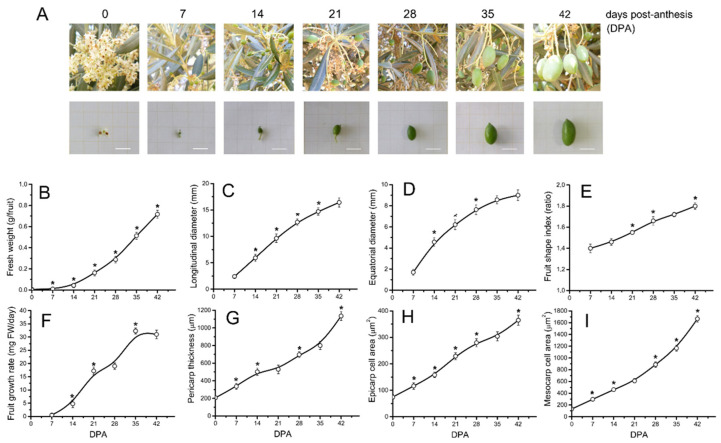
Growth of ‘Picual’ olive fruits. (**A**) Morphological changes of olive fruit during early fruit development. Increase in ‘olive fruit length and diameter as a function of days post anthesis (DPA). (**B**) Changes in fresh weight (FW) (g fruit^−1^), (**C**) longitudinal diameter (mm), (**D**) transverse diameter (mm), (**E**) fruit-shape index, (**F**) growth rate (mg FW day^−1^), (**G**) pericarp thickness (mm), (**H**) epicarp cell size (µm^2^), and (**I**) mesocarp cell size (µm^2^) of developing olive fruit at 0, 7, 14, 21, 28, 35, and 42 DPA. Fruit-shape index is length-to-width ratio of the fruit. The cell area of fruit mesocarp and epicarp cells was measured during early fruit development (staining was with Calcofluor White) using confocal microscopy. Asterisks indicate statistically significant changes based on an unpaired Student’s *t*-test (*p* < 0.05) from the preceding point. Data are means ± SE (n ≥ 20). Scale bar 10 mm. DPA: Days Post-Anthesis.

**Figure 2 ijms-24-00961-f002:**
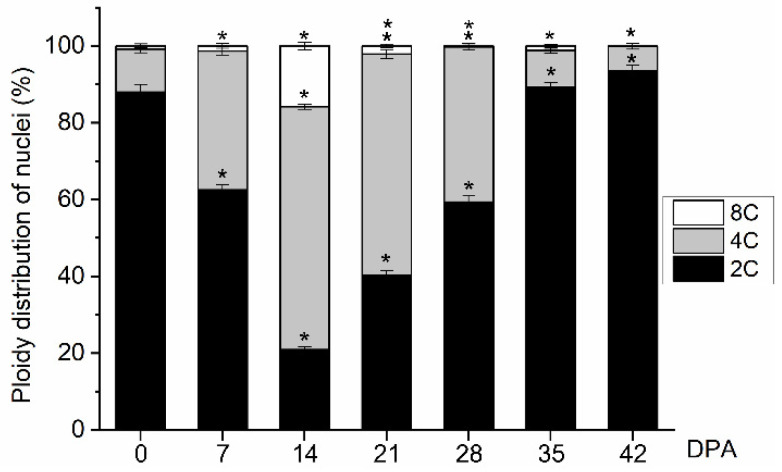
Nuclear ploidy levels from olive fruits during early development. The percentage of nuclei in 2C, 4C, and 8C are shown from 0 to 42 DPA in the developing fruits. Each point is the average of four samples. Asterisks denote significant differences based on unpaired Student’s *t*-test (*p* < 0.05) from the preceding point and bars are ± SE.

**Figure 3 ijms-24-00961-f003:**
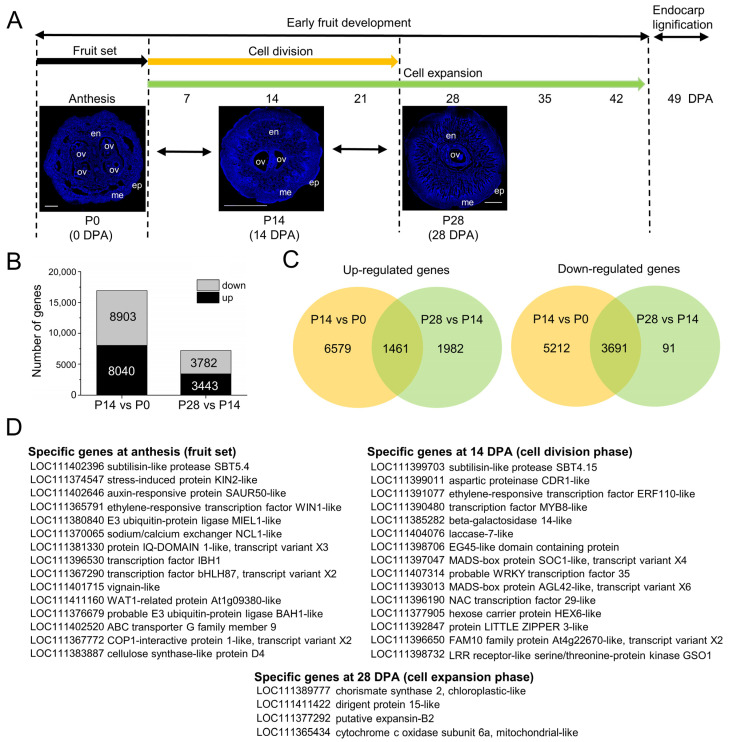
Design of RNA-seq analysis and olive fruit genes during early fruit development. (**A**) Cross-sections of ovaries at anthesis stage (P0) and developing fruits at 14 DPA (P14) and 28 DPA (P28) used in RNA-seq analysis. Micrographs showing changes in fruit surface during early fruit development, stained with Calcofluor White. Calcofluor White staining of cellulose indicated the cell wall; en, endocarp; ep, epicarp; me, mesocarp; ov, ovule. Scale bar 50 µm in P0, and 500 µm in P14 and P28. (**B**) Distribution of genes differentially expressed during early fruit development (P14 vs. P0, and P28 vs. P14 comparisons). (**C**) Overlap of upregulated and downregulated fruit genes during early olive fruit development. This Figure shows that 1461 were upregulated in both comparisons and that 3691 were downregulated in both comparisons. (**D**) Specific genes differentially expressed during early fruit development. Top 15 genes at anthesis (P0), and at 14 DPA (P14). Additional information is presented in Appendix A.

**Figure 4 ijms-24-00961-f004:**
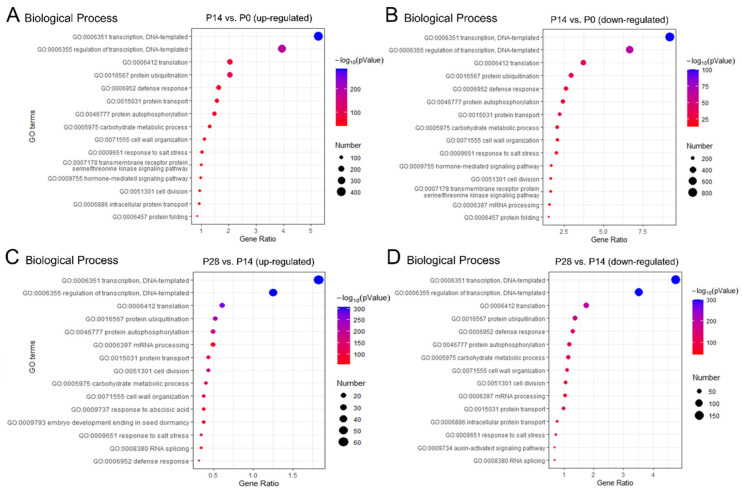
Functional analysis of DEGs during early olive fruit development. The enrichment analysis of GO ‘biological process’ terms based on DEGs in the olive flowers at anthesis stage (P0), and the developing fruits at 14 (P14) and 28 (P28) DPA. (Top 15). (**A**) Bubble Plot of GO ‘biological process’ terms in the GO annotations of the genes of the 8040 transcripts with increased transcript accumulation, (**B**) and of the genes of the 8903 transcripts with decreased transcript accumulation in the P14 vs. P0 comparison; (**C**) Bubble Plot of GO ‘biological process’ terms in the GO annotations of the genes of the 3443 transcripts with increased transcript accumulation, and (**D**) of 3782 transcripts with decreased transcript accumulation in the P28 vs. P14 comparison. The *Y*-axis and *X*-axis denote GO name and gene ratio, respectively. The color of each bubble represents −log_10_ (*p*-value), and each bubble size represents the count of DEGs. Additional information is presented in Appendix A.

**Figure 5 ijms-24-00961-f005:**
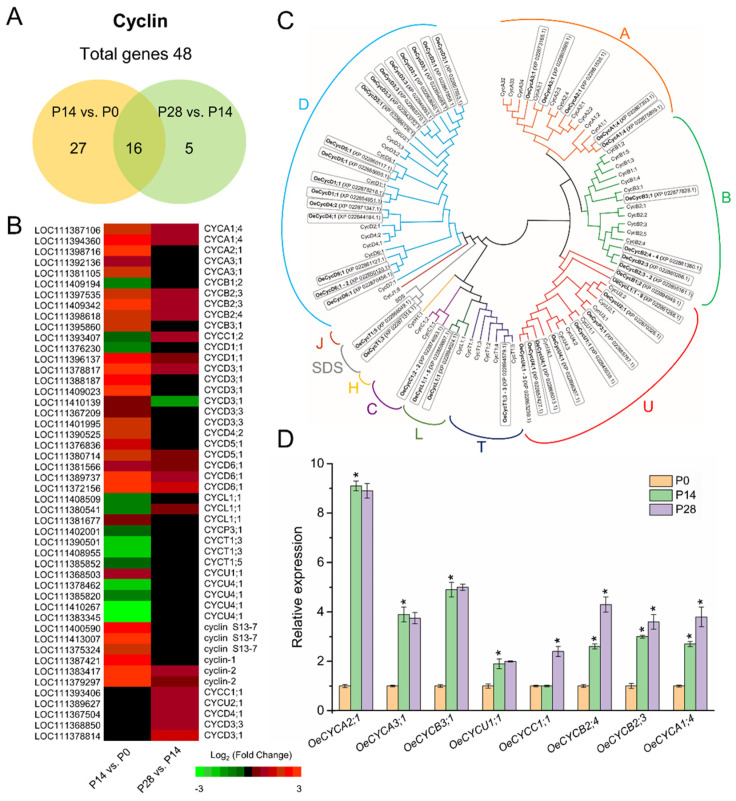
Differential gene expression of *CYC* genes during early olive fruit development. (**A**) Venn diagram showing numbers of overlapping *CYC* genes in both the fruit at 14 DPA (P14) versus fruit at 0 DPA (P0), and in the fruit at 28 DPA (P28) versus fruit at 14 DPA (P14) comparisons. (**B**) Expression values are represented in a heatmap as Log_2_ Fold Change in the P14 versus P0 comparison, and the P28 versus P14 comparison, and the color key is indicated at the bottom. (**C**) Phylogenetic analysis of olive *CYC* with other *CYC* genes. The sequences included in this alignment are from olive (*Olea europaea* var. sylvestris Annotation Report; https://www.ncbi.nlm.nih.gov/genome/annotation euk/Olea europaea var. sylvestris/100/, accessed on 10 January 2020), and Arabidopsis (http://www.arabidopsis.org/, accessed on 10 January 2022). The CYC proteins studied from the present work are enclosed in an open box. (**D**) Expression of 8 selected CYC genes in olive fruits at 0 (P0), 14 (P14), and 28 (P28) DPA. Analysis of transcript levels of genes by qRT-PCR. Genes and their primers are shown in Appendix A. Relative expression values were normalized to the lowest expression value taken as 1. The data represent the mean values (±SEs) of duplicate experiments from three independent biological samples. Statistical significance compared with the preceding point was determined using Student’s *t-*test. * *p* < 0.05. Additional information on the *CYC* genes is presented in Appendix A.

**Figure 6 ijms-24-00961-f006:**
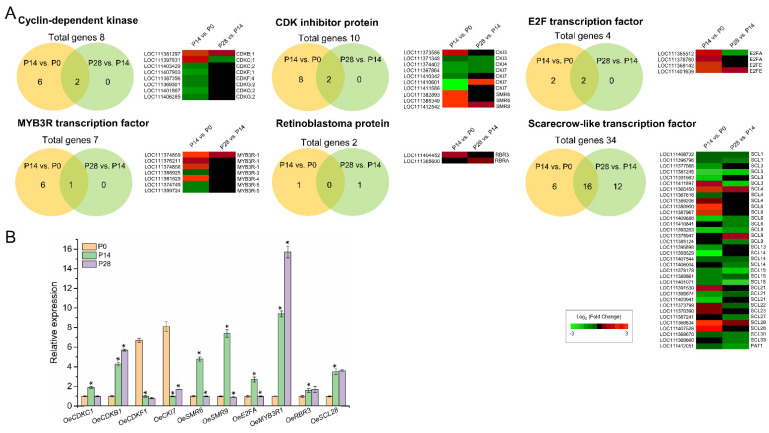
Differential gene expression of candidate cell cycle-related genes during early olive fruit development. (**A**) Venn diagram representing overlap between differential expressed genes *CDK, CKI, E2F, MYB3R, RBR*, and *SCL* in both the comparison of fruit at 14 DPA (P14) versus fruit at 0 DPA (P0), and the comparison of fruit at 28 DPA (P28) versus fruit at 14 DPA (P14). Expression values are represented in a heatmap as Log_2_ Fold Change in the P14 versus P0, and the P28 versus P14 comparisons, and the color key is indicated at the bottom. (**B**) Expression of cell cycle-related genes in olive fruits at 0 (P0), 14 (P14), and 28 (P28) DPA. Analysis of transcript levels of cell cycle-related genes by qRT-PCR. Genes and their primers are shown in Appendix A. Relative expression values were normalized to the lowest expression value taken as 1. The data represent the mean values (±SEs) of duplicate experiments from three independent biological samples. Statistical significance compared with the preceding point was determined using Student’s *t-*test. * *p* < 0.05. Additional information on the cell cycle-related genes is presented in Appendix A.

**Figure 7 ijms-24-00961-f007:**
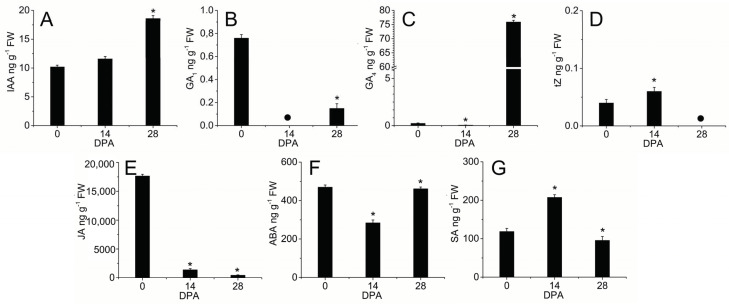
Profiles of (**A**) indole-3-acetic acid (IAA), (**B**) gibberellin 1 (GA_1_), (**C**) gibberellin 4 (GA_4_), (**D**) *trans*-Zeatin (*t*Z), (**E**) jasmonic acid (JA), (**F**) abscisic acid (ABA), and (**G**) salicylic acid (SA) levels in olive fruits at 0 (P0), 14 (P14), and 28 (P28) DPA during early fruit development. Hormone levels not detected are indicated by a black dot (•). Data are the means ± SD of three biological replicates with three technical repeats each. Statistically significant differences based on unpaired Student’s *t-*test at *p* < 0.05 are denoted by an asterisk.

**Figure 8 ijms-24-00961-f008:**
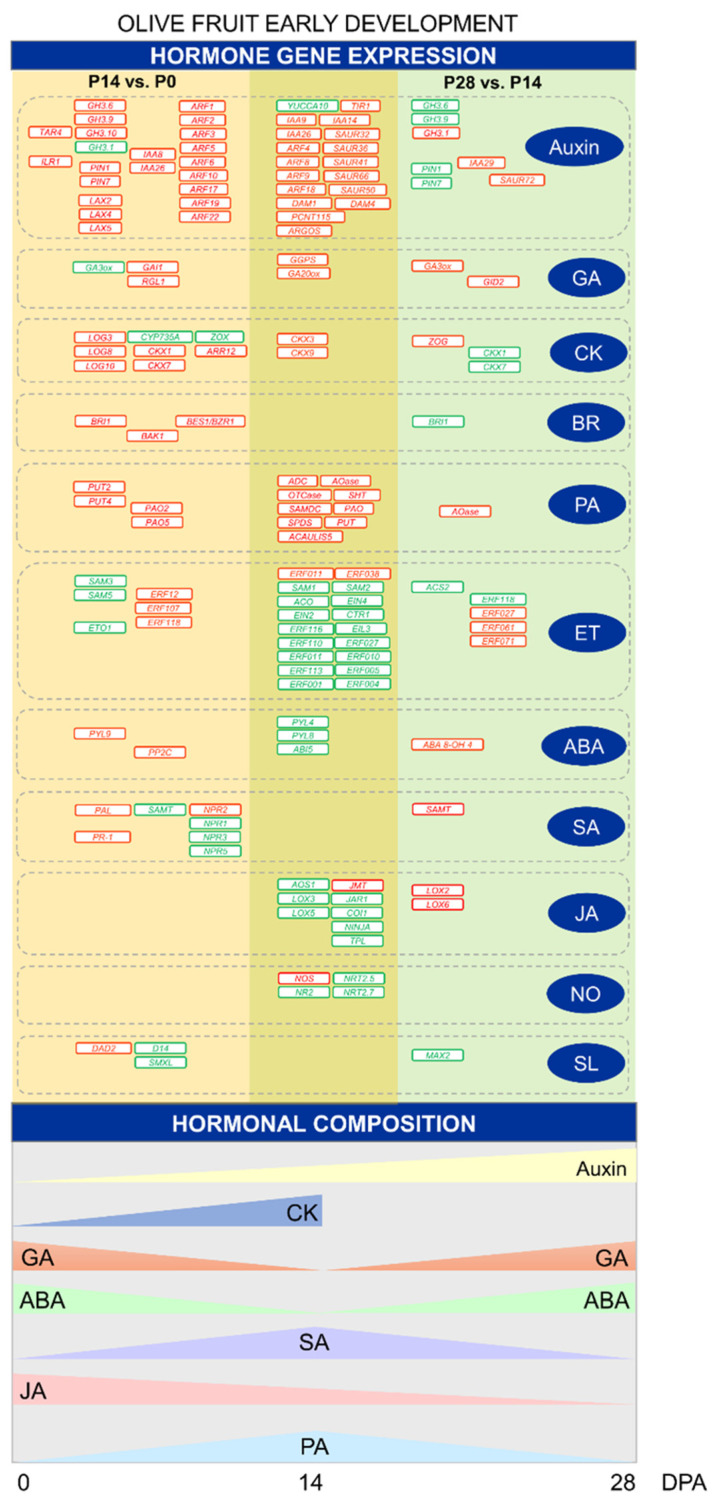
Summary figure including hormonal composition and the most important hormone-related DEGs during early fruit development in olive. DEGs in red represent genes that increased their expression and DEGs in green represent genes that decreased their expression. The middle column of the figure represents DEGs in common in both comparisons. Additional information on the hormone-related genes is presented in Appendix A.

**Figure 9 ijms-24-00961-f009:**
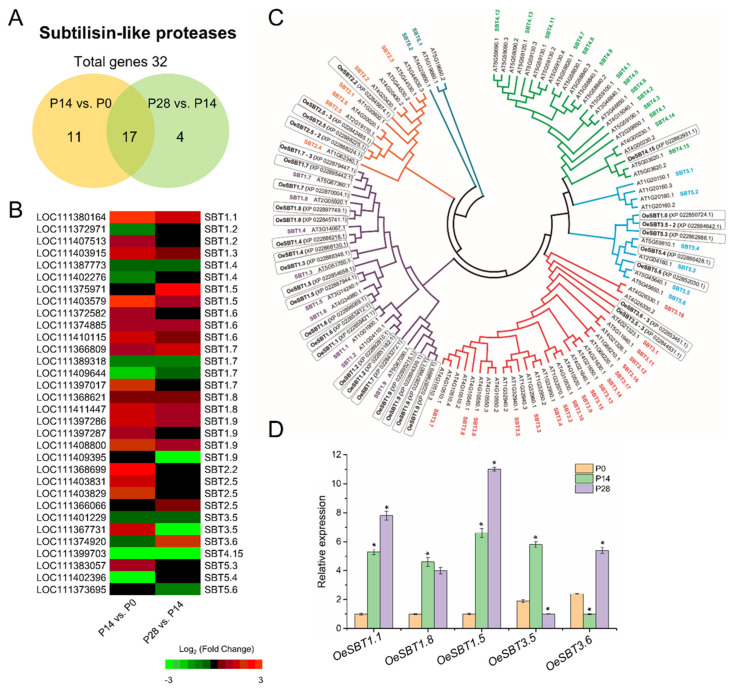
Differential gene expression of subtilisin-like protease (*SBT*) genes during early olive fruit development. (**A**) Venn diagram showing numbers of overlapping *SBT* genes in both the fruit at 14 DPA (P14) versus fruit at 0 DPA (P0), and in the fruit at 28 DPA (P28) versus fruit at 14 DPA (P14) comparisons. (**B**) Expression values are represented in a heatmap as Log_2_ Fold Change in the P14 versus P0 comparison, as well as the P28 versus P14 comparison; the color key is indicated at the bottom. (**C**) Phylogenetic analysis of olive SBT (OeSBT) with other SBT genes. The sequences included in this alignment are from olive (*Olea europaea* var. sylvestris Annotation Report; https://www.ncbi.nlm.nih.gov/genome/annotation euk/Olea europaea var. sylvestris/100/, accessed on 10 January 2020) and Arabidopsis (http://www.arabidopsis.org/, accessed on 10 January 2022). The SBT proteins studied from the present work are enclosed in an open box. (**D**) Expression of five selected *SBT* genes in olive fruits at 0 (P0), 14 (P14), and 28 (P28) DPA. Analysis of transcript levels of genes by qRT-PCR. Genes and their primers are shown in Appendix A. Relative expression values were normalized to the lowest expression value taken as 1. The data represent the mean values (±SEs) of duplicate experiments from three independent biological samples. Statistical significance compared with the preceding point was determined using a two-sided Student’s *t-*test. * *p* < 0.05. Additional information on *SBT* genes is presented in Appendix A.

**Figure 10 ijms-24-00961-f010:**
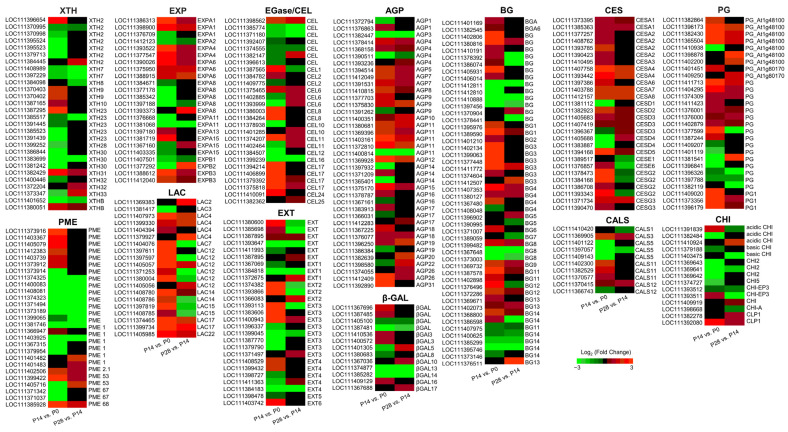
Expression profile of family genes encoding various cell wall proteins during early fruit development in olive. Expression values are represented in a heatmap as Log_2_ Fold Change in both the P14 versus P0 (P14 vs. P0) and the P28 versus P14 (P28 vs. P14) comparisons, and the color key is indicated at the bottom. Additional information on the cell wall-related genes is presented in Appendix A.

**Figure 11 ijms-24-00961-f011:**
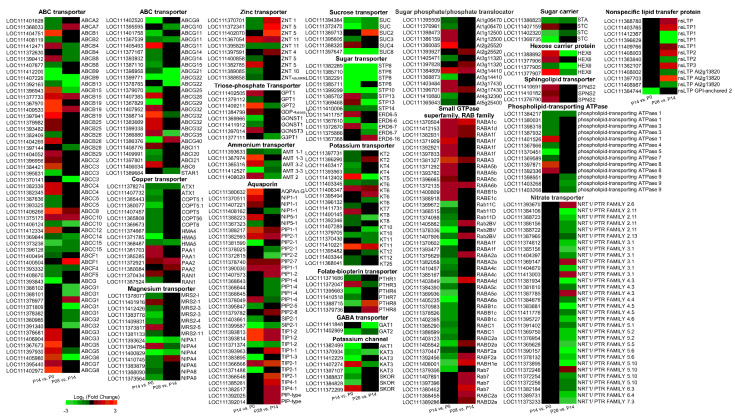
Expression profile of transport-related genes during early fruit development in olive. Expression values are represented in a heatmap as Log_2_ Fold Change in both the P14 versus P0 and P28 versus P14 comparisons; the color key is indicated at the bottom. Additional information on the transport-related genes is presented in Appendix A.

**Figure 12 ijms-24-00961-f012:**
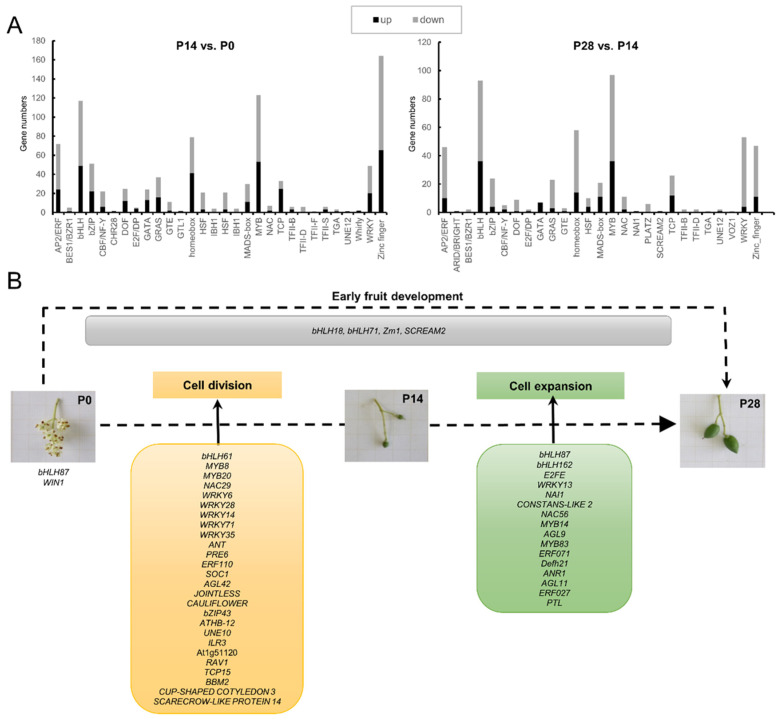
Olive transcription factors (TFs) induced or repressed during early fruit development. (**A**) Summary of the number of significant changes in TF transcripts between the different families during early fruit development in olive. Comparisons of fruit at 14 versus 0 DPA (P14 vs. P0), and 28 versus 14 DPA (P28 vs. P14) for significantly upregulated (black) and downregulated (grey) transcripts revealed differences in the families of TFs during early olive fruit development. (**B**) The TFs involved in regulating cell division and expansion phases during early fruit development in olive. The TFs (log_2_FoldChange > 3, *p* < 0.01) were chosen for candidate TFs between the samples.

## Data Availability

The results of this study are available on request from the corresponding author (Maria C. Gomez-Jimenez).

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
