# Peer review of "Characterization of Transcriptome Dynamics during Early Fruit Development in Olive (Olea europaea L.)"

_ijms, 2023, doi:10.3390/ijms24020961_

Round 1

Reviewer 1 Report

 In my opinion the experimental design is very appropriate and the results of great interest.

The  study shows a complete analysis of gene expression and hormonal control during early fruit development in olive. The authors analyse the cytology, ploidy, and transcriptome dynamics associated with early fruit development.

1 Analysis of the mitotic activity in the developing olive fruits to characterize the duration of the cell division.

2 Transcriptional and hormonal changes in olive fruit were investigated during early fruit development 

3 Candidate genes associated with distinct phases of early  development.

3 Potential genes associated  to hormonal pathways were determined to participate in the phases of cell division and expansion during olive fruit early development

Author Response

Response to Reviewer 1 Comments

In my opinion the experimental design is very appropriate and the results of great interest.

The  study shows a complete analysis of gene expression and hormonal control during early fruit development in olive. The authors analyse the cytology, ploidy, and transcriptome dynamics associated with early fruit development.

1 Analysis of the mitotic activity in the developing olive fruits to characterize the duration of the cell division.

2 Transcriptional and hormonal changes in olive fruit were investigated during early fruit development 

3 Candidate genes associated with distinct phases of early  development.

3 Potential genes associated  to hormonal pathways were determined to participate in the phases of cell division and expansion during olive fruit early development

Response: We are very grateful to the Reviewer 1 for the comments on our manuscript.

Reviewer 2 Report

This manuscript is a powerful article with relevant information about the development of the fruit and the main genes involved in its cell division and cell expansion.

The presentation is very good, with quality images and quite clarifying figures. However, there are some issues and points that have to be improved so that the article can be published in my opinion.

Introduction

It is written in a messy way. The authors begin by talking about the stages of fruit development and then describe in a paragraph the stage of fruit maturation with the formation of lipids and oil accumulation after stone formation. Once this stage of maturation has been described, the authors return to write about the stage of formation of the immature fruit.

There are some sentences in which the authors should be more explicit when making certain statements (see the attached file where some comments on the matter are indicated)

On the other hand, the authors must suppress the wording using the "we" in the manuscript, changing this wording in the impersonal sense that should be used for this type of article.

Methods:

Samples used for transcriptomics: More detail about the pool of samples  made for RNA extraction (see attached file with comments)

Reference genome: A wild olive genome has been used for the alignment when there are already publications of cultivated olive genomes of the same variety that they have used (Picual). Why?

Hormone measurement:

At least indicate which quantification methodology has been used to measure the hormones and which hormones have been quantified

Results:

transcriptomic analysis

Why hasn't the comparison between 0DPA vs 28DPA been made? I believe that there are genes that may be exclusive to this stage and that, just as the other two stages have been compared, 0DPA vs 14DPA and 14 DPA vs 28 DPA, these phases of development 0 DPAvs 28DPA should also be compared. I suggest that this comparison be made

The figures on many occasions have a very small font size so that many of the names indicated in them are not visible. I suggest increasing the font size on some of the schemes

Hormones and their measurement

I think more information should be given about the measurement methodology used to measure the hormones indicated. I also suggest that in the interpretation of the transcriptome, and specifically in relation to the gene expression of the multitude of genes that are related to the different hormones, the general trend of each group of hormones in each phase of fruit development is explained because It is not understood nor can we reach specific conclusions about the expression of the different groups of hormones

supplementary material

Figures s5 to s21 are too speculative as they are represented since in many of the processes shown in the schemes there are reactions and processes that are currently unknown and it is not yet known exactly what the signaling pathway is in some of the schemes represent

I suggest that if you want to keep all of them, which I find very attractive, put them at the bottom of the figure (for example, in the case of figure S5: "possible signalling pathway suggested by the authors to explain the function of genes involved in oxidative phosphorylation")

The captions to the figures in the supplementary material cannot remain as they appear, because there are many signalling pathways currently unknown and many genes indicated in those pathways whose function is unknown.

I attach a file with all the suggestions indicated in the text

Author Response

Response to comments by Reviewer 2

This manuscript is a powerful article with relevant information about the development of the fruit and the main genes involved in its cell division and cell expansion.

The presentation is very good, with quality images and quite clarifying figures. However, there are some issues and points that have to be improved so that the article can be published in my opinion.

Point 1: Introduction

It is written in a messy way. The authors begin by talking about the stages of fruit development and then describe in a paragraph the stage of fruit maturation with the formation of lipids and oil accumulation after stone formation. Once this stage of maturation has been described, the authors return to write about the stage of formation of the immature fruit.

Response 1: This been rewritten as suggested.

“Because of in the increasing oil content during fruit pulp growth, which can reach 30% (fresh weight) at full ripening [6], and due to the high commercial value of the oil, most studies to date have attempted to elucidate the molecular bases of olive fruit ripening [7-20], but, to date, few studies have shown candidate genes associated with early fruit development in olive [21-28]. In particular, the transcriptomic control of changes associated with early fruit development in olive remains unknown.”

Point 2: There are some sentences in which the authors should be more explicit when making certain statements (see the attached file where some comments on the matter are indicated)

Response 2: These sentences have been made more specific in the new version.

Point 3: On the other hand, the authors must suppress the wording using the "we" in the manuscript, changing this wording in the impersonal sense that should be used for this type of article.

Response 3: The writing has been modified to avoid the first person, as suggested.

“Moreover, exogenous brassinosteroid (BR) application promoted the early fruit growth in olive, whereas the blocking of BR synthesis with brassinazole (Brz) slowed down the fruit growth rate [62]. Likewise, the data have provided new findings on the role of BRs in modulating the composition and gene expression of sterols and sphingolipids, these being lipophilic membrane components essential for cellular functions during early olive fruit growth [62,63]. This latter study also revealed the up-regulation of ?-sitosterol biosynthesis by BR at the transcriptional level during early olive fruit growth. In addition, previous data have demonstrated that free endogenous PAs may regulate olive flower anthesis mainly through S-adenosyl methionine decarboxylase (SAMDC) enzyme activity and expression gene localized in the ovary; the same study has indicated that PAs appear to correlate positively with cell division during early fruit development in olive [27].”

“For this, the cytology, ploidy, and transcriptome dynamics associated with early fruit development in combination with data on hormonal content were analysed.”

Point 4: Methods:

Samples used for transcriptomics: More detail about the pool of samples made for RNA extraction (see attached file with comments)

Response 4: This point has been clarified as suggested in the new version.

Point 5: Reference genome: A wild olive genome has been used for the alignment when there are already publications of cultivated olive genomes of the same variety that they have used (Picual). Why?

Response 5: Our RNA-seq analysis was conducted in November 2019. At that time, we used the genome Olea europaea var. Sylvestris (one of the olive genomes available) for the RNA-seq analysis. As a consequence of the COVID pandemic (beginning in early 2020), technical problems with the real-time PCR equipment, and the sick leave of our doctoral student (M.C. Camarero) for the year 2021, the research was considerably delayed. Meanwhile, the genome of this cultivar (‘Picual’) became available.

Although new olive genomes are available, the genome used meets the needs of the present work, and the genes and the protein domains evaluated have been annotated, guaranteeing of the veracity of the study. While we consider the reviewer’s observation to be valid, we believe that the use of the genome of the Picual cultivar for the RNA-seq analysis could be a further step forward, and therefore a subsequent work. In future research, we intend to map our NGS sequencing results against the genome of Olea europaea L. subsp. europaea (https://denovo.cnag.cat/olive; Julca et al., 2020) and against the genome of Olea europaea cv. ‘Picual’ (Jiménez-Ruiz et al., 2020).

Jiménez-Ruiz et al. 2020. Plant Genome, e20010.

Julca et al. 2020. BMC Biol. 18(1):148.

Point 6: Hormone measurement:

At least indicate which quantification methodology has been used to measure the hormones and which hormones have been quantified.

Response 6: This information has been provided as recommended by the reviewer.

Point 7: Results:

transcriptomic analysis

Why hasn't the comparison between 0DPA vs 28DPA been made? I believe that there are genes that may be exclusive to this stage and that, just as the other two stages have been compared, 0DPA vs 14DPA and 14 DPA vs 28 DPA, these phases of development 0 DPAvs 28DPA should also be compared. I suggest that this comparison be made

Response 7: Despite the global analysis of gene expression, the goal of this work was the sequential identification of genes linked with cell division and expansion phases in early developing olive fruit. The use of a Venn diagram (https://bioinfogp.cnb.csic.es/tools/venny/) provided a specific gene expression for each comparison, and a large number of specific genes at 0, 14, and 28 DPA were identified in the present study (Table S8; Figure 3D), and, hence, the coverage has been sufficient for the goals of the work. Thus, they were considered to be main candidates for further molecular research on the programmes of cell division and expansion in olive fruit and their regulation. This study provides the first thorough analysis available for a complete picture of gene expression during early fruit development in olive. While we consider the reviewer's observations to be accurate, we believe that the comparison between 0DPA vs 28DPA for the RNA-seq analysis in different olive cultivars (with different final fruit size) could constitute a valuable subsequent work.

At the moment, we are preparing a manuscript on the comparison between 0 DPA and 28 DPA for RNA-seq analyses of several olive cultivars of different final fruit size (including ‘Picual’) and consequently the comparison 0 DPA and 28 DPA of this cultivar will be presented in the forthcoming work.

Point 8: The figures on many occasions have a very small font size so that many of the names indicated in them are not visible. I suggest increasing the font size on some of the schemes.

Response 8: The changes have been made as suggested, enlarging the font size in Figures 3, 10, 11, and S22.

Point 9: Hormones and their measurement

I think more information should be given about the measurement methodology used to measure the hormones indicated.

Response 9: Additional information has been provided. Comments have been added in the “Material and Methods” section for clarification.

Point 10: I also suggest that in the interpretation of the transcriptome, and specifically in relation to the gene expression of the multitude of genes that are related to the different hormones, the general trend of each group of hormones in each phase of fruit development is explained because It is not understood nor can we reach specific conclusions about the expression of the different groups of hormones

Response 10: The general trend of each group of hormones in each phase of fruit development has been explained, as suggested.

Point 11: supplementary material

Figures s5 to s21 are too speculative as they are represented since in many of the processes shown in the schemes there are reactions and processes that are currently unknown and it is not yet known exactly what the signaling pathway is in some of the schemes represent

I suggest that if you want to keep all of them, which I find very attractive, put them at the bottom of the figure (for example, in the case of figure S5: "possible signalling pathway suggested by the authors to explain the function of genes involved in oxidative phosphorylation")

The captions to the figures in the supplementary material cannot remain as they appear, because there are many signalling pathways currently unknown and many genes indicated in those pathways whose function is unknown.

Response 11: We agree with the reviewer, and therefore we have removed Figures S5-S21 from the supplementary material and the manuscript in the new version.

Point 12:

I attach a file with all the suggestions indicated in the text

Response 12: All the suggestions have been carefully addressed and the manuscript has been revised as suggested.

Based on our cytological and ploidy analyses of early olive fruit development, particularly evident is the transition between 14 and 28 DPA, which corresponds to the shift from intense cell division to cell-division arrest in ‘Picual’ olive fruit. From 21 to 28 DPA, dramatic pericarp cell expansion occurs with a cell-expansion rate (cell area/day) doubling up to 42 DPA, whereas pericarp cell expansion occurs at similar rates for 21 DPA. Although the maximum relative rate of cell expansion in the fruit pericarp occurs at 42 DPA, at this stage, the endocarp lignification has started and is close to completion (49 DPA). Thus, in the present study, we selected the fruit sample at 28 DPA (P28) which represents the onset of intense cell-expansion phase for the identification of post-mitotic cell-expansion-related genes.

In Figure 7, the CK (tZ) levels not detected in fruit at 28 DPA are indicated by a black dot (•). In Figure 8, the CK (tZ) levels are not detected in fruit at 28 DPA.

Point 13: supplementary material. Figure S1: Missing comparison between 0 DPA vs 28 DPA

Response 13: This information has been provided as suggested.

At the moment, we are preparing a manuscript on the comparison between 0 DPA and 28 DPA for RNA-seq analyses of several olive cultivars of different final fruit size (including ‘Picual’) and consequently the comparison 0 DPA and 28 DPA of this cultivar will be presented in the forthcoming work.

Point 14: supplementary material. Figure S22: Increase the font size please. You can not see the name of the genes and the samples that you are comparing. Revise this detail in all the graphics and figures.

Response 14: The changes have been made as suggested, enlarging the font size in Figures 3, 10, 11, and S22.

We are very grateful to the Reviewer 2 for the constructive comments on our manuscript and we hope we have adequately addressed the concerns in the prepapation of this new version of the manuscript.

Round 2

Reviewer 2 Report

Dear authors: I believe this manuscript has scientific rigour and it is a good paper that could be improved in some aspects. For example, FROM MY POINT OF VIEW IT IS ESSENTIAL TO MAKE A COMPARISON BETWEEN THE FRUIT SAMPLES 0 days post anthesis and 28 days post anthesis. Given the comments on future papers in which these comparisons are provided, I consider that the manuscript can be published according to the second version that you send me. I also consider it important to eliminate the supplementary schemes on signalling routes that are too speculative. Merry Christmas